# Double Neural Counterfactual Regret Minimization

**Hui Li**[§*]**, Kailiang Hu**[§]**, Shaohua Zhang**[§]**, Yuan Qi**[§]**, Le Song**[§♮]
[§]Ant Financial Services Group [♮]Georgia Institute of Technology
{ken.lh, hkl163251, yaohua.zsh, yuan.qi, le.song}@antfin.com
lsong@cc.gatech.edu

## Abstract

Counterfactual regret minimization (CFR) is a fundamental and effective technique for solving Imperfect Information Games (IIG). However, the original CFR algorithm only works for discrete states and action spaces, and the resulting strategy is maintained as a tabular representation. Such tabular representation limits the method from being directly applied to large games. In this paper, we propose a double neural representation for the IIGs, where one neural network represents the cumulative regret, and the other represents the average strategy. Such neural representations allow us to avoid manual game abstraction and carry out end-to-end optimization. To make the learning efficient, we also developed several novel techniques including a robust sampling method and a mini-batch Monte Carlo Counterfactual Regret Minimization (MCCFR) method, which may be of independent interests. Empirically, on games tractable to tabular approaches, neural strategies trained with our algorithm converge comparably to their tabular counterparts, and significantly outperform those based on deep reinforcement learning. On extremely large games with billions of decision nodes, our approach achieved strong performance while using hundreds of times less memory than the tabular CFR. On head-to-head matches of hands-up no-limit texas hold'em, our neural agent beat the strong agent ABS-CFR [1] by $9.8\pm4.1$ chips per game. It's a successful application of neural CFR in large games.

## 1 Introduction

While significant advance has been made in addressing large perfect information games, such as Go (Silver et al., 2016), solving imperfect information games remains a challenging task. For Imperfect Information Games (IIG), a player has only partial knowledge about her opponents before making a decision, so that she has to reason under the uncertainty about her opponents' information while exploiting the opponents' uncertainty about herself. Thus, IIGs provide more realistic modeling than perfect information games for many real-world applications, such as trading, traffic routing, and politics.

Nash equilibrium is a typical solution concept for a two-player perfect-recall IIG. One of the most effective approaches is CFR (Zinkevich et al., 2007), which minimizes the overall counterfactual regret so that the average strategies converge to a Nash equilibrium. However the original CFR only works for discrete states and action spaces, and the resulting strategy is maintained as a tabular representation. Such tabular representation limits the method from being directly applied to large games. To tackle this challenge, one can simplify the game by grouping similar states together to solve the simplified (abstracted) game approximately via tabular CFR (Zinkevich et al., 2007; Lanctot et al., 2009). Constructing an effective abstraction, however, demands rich domain knowledge and its solution may be a coarse approximation of true equilibrium.

Function approximation can be used to replace the tabular representation. Waugh et al. (2015) combines regression tree function approximation with CFR based on handcrafted features, which is called Regression CFR (RCFR). However, since RCFR uses full traversals of the game tree, it is still impractical for large games. Moravcik et al. (2017) propose a seminal approach DeepStack, which uses fully connected neural networks to represent players' counterfactual values, tabular CFR however was used in the subgame solving. Jin et al. (2017) use deep reinforcement learning to solve regret minimization problem for single-agent settings, which is different from two-player perfect-recall IIGs.

---

[*]The previous versions of this work were published in AAAI 2019 workshop on RLG and ICML 2019 workshop on RWSDM. It takes more than one year for us to reimplement DeepStack and evaluate our method on large-scale heads-up no-limit Texas Hold'em. Correspondence to: Hui Li, lihuiknight@google.com.

[1] ABS-CFR is the advanced version of HITSZ_LMW_2pn, who won the third prize of the 2018 Annual Computer Poker Competition (ACPC). In our experiment, we chosen its advanced version as the benchmark.

To learn approximate Nash equilibrium for IIGs in an end-to-end manner, Heinrich et al. (2015) and Heinrich & Silver (2016) propose eXtensive-form Fictitious Play (XFP) and Neural Fictitious Self-Play (NFSP), respectively, based on deep reinforcement learning. In a NFSP model, the neural strategies are updated by selecting the best responses to their opponents' average strategies. These approaches are advantageous in the sense that they do not rely on abstracting the game, and accordingly their strategies can improve continuously with more optimization iterations. However fictitious play empirically converges much slower than CFR-based approaches. Srinivasan et al. (2018) use actor-critic policy optimization methods to minimize regret and achieve performance comparable to NFSP.

Thus it remains an open question whether a purely neural-based end-to-end approach can achieve comparable performance to tabular based CFR approach. In the paper, we solve this open question by designing a double neural counterfactual regret minimization (DNCFR) algorithm [2]. To make a neural representation, we modeled imperfect information game by a novel recurrent neural network with attention. Furthermore, in order to improve the convergence of the neural algorithm, we also developed a new sampling technique which converged much more efficient than the outcome sampling, while being more memory efficient than the external sampling. In the experiment, we conducted a set of ablation studies related to each novelty. The experiments showed DNCRF converged to comparable results produced by its tabular counterpart while performing much better than NFSP. In addition, we tested DNCFR on extremely large game, heads-up no-limit Texas Hold'em (HUNL). The experiments showed that DNCFR with only a few number of parameters achieved strong neural strategy and beat ABS-CFR.

## 2 BACKGROUND

● **Notation.** Figure 1 illustrates an extensive game for a finite set $N = \{0, 1, ..., n-1\}$ of $n$ players. Define $x_i^v$ as the **hidden information** of player $i$ in IIG. $x_{-i}^v$ refers to hidden variables of all players other than $i$. $H$ refers to a finite set of histories. $h \in H$ denotes a possible **history** (or state), which consists of each player's hidden variable and actions taken by all players including chance. The empty sequence $\emptyset$ is a member of $H$. $h_j \sqsubseteq h$ denotes $h_j$ is a prefix of $h$. $Z \subseteq H$ denotes the

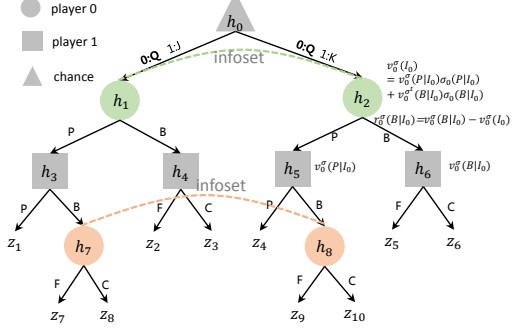

Figure 1: Extensive-Form IIG and Information Set

terminal histories and any member $z \in Z$ is not a prefix of any other sequences. $A(h) = \{a : ha \in H\}$ is the set of available actions after non-terminal history $h \in H \setminus Z$. A **player function** $P$ assigns a member of $N \cup \{c\}$ to each non-terminal history, where $c$ is the chance ( we set $c = -1$). $P(h)$ is the player who takes an action after history $h$. For each player $i$, imperfect information is denoted by **information set (infoset)** $I_i$. All states $h \in I_i$ are indistinguishable to $i$. $\mathcal{I}_i$ refers to the set of infosets of $i$. The **utility function** $u_i(z)$ defines the payoff of $i$ at state $z$. See appendix B.1 for more details.

A **strategy profile** $\sigma = \{\sigma_i | \sigma_i \in \Sigma_i, i \in N\}$ is a collection of strategies for all players, where $\Sigma_i$ is the set of all possible strategies for player $i$. $\sigma_{-i}$ refers to strategy of all players other than player $i$. For play $i \in N$, the **strategy** $\sigma_i(I_i)$ is a function, which assigns an action distribution over $A(I_i)$ to infoset $I_i$. $\sigma_i(a|h)$ denotes the probability of action $a$ taken by player $i$ at state $h$. In IIG, $\forall h_1, h_2 \in I_i$ , we have $\sigma_i(I_i) = \sigma_i(h_1) = \sigma_i(h_2)$. For iterative method such as CFR, $\sigma^t$ refers to the strategy profile at $t$-th iteration. The **state reach probability** of history $h$ is denoted by $\pi^\sigma(h)$ if players take actions according to $\sigma$. The reach probability is also called **range** in DeepStack (Moravcik et al., 2017). Similarly, $\pi_i^\sigma(h)$ refers to those for player $i$ while $\pi_{-i}^\sigma(h)$ refers to those for other players except for $i$. For an empty sequence $\pi^\sigma(\emptyset) = 1$. One can also show that the reach probability of the opponent is proportional to posterior

---

**Algorithm 1:** CFR Algorithm

**For** $t = 1$ *to* $T$ **do**

$$v_i^{\sigma^t}(I_i) = \sum_{h \in I_i, h \sqsubseteq z, z \in Z} \pi_i^{\sigma^t}(h, z) \pi_{-i}^{\sigma^t}(z) u_i(z). \quad (1)$$

$$r_i^{\sigma^t}(a|I_i) = v_i^{\sigma^t}(a|I_i) - v_i^{\sigma^t}(I_i). \quad (2)$$

$$R_i^t(a|I_i) = R_i^{t-1}(a|I_i) + r_i^{\sigma^t}(a|I_i). \quad (3)$$

$$\sigma_i^{t+1}(a|I_i) = \begin{cases} \frac{1}{|A(I_i)|} & \text{if } \sum_{a \in A(I_i)} R_i^{t,+}(a|I_i) = 0 \\ \frac{R_i^{t,+}(a|I_i)}{\sum_{a \in A(I_i)} R_i^{t,+}(a|I_i)} & \text{otherwise.} \end{cases}$$

$$\quad (4)$$

$$S^t(a|I_i) = S^{t-1}(a|I_i) + \pi_i^{\sigma^t}(I_i)\sigma_i^t(a|I_i). \quad (5)$$

$$\bar{\sigma}_i^T(a|I_i) = \frac{S^T(a|I_i)}{\sum_{a \in A(I_i)} S^T(a|I_i)}. \quad (6)$$

---

[2] Solving IIGs via function approximation methods is an important and challenging problem. In the past year, several concurrent works (Lockhart et al., 2019; Brown et al., 2018; Steinberger, 2019) have been proposed to address this problem. We will discuss their differences in Section 6.

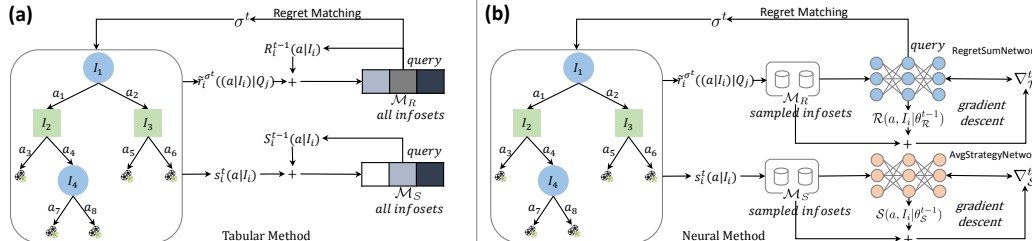

Figure 2: (a) tabular CFR and (b) our double neural CFR framework. $\tilde{r}_i^{\sigma^t}((a|I_i)|Q_j)$ is the estimated regret in MCCFR, $R_i^{t-1}(a|I_i)$ is the cumulative regret, $s_i^t(a|I_i)$ is the weighted additional strategy and $S_i^{t-1}(a|I_i)$ is the cumulative behavior strategy. In tabular CFR, cumulative regret and strategy are stored in the tabular memory, which limits it to solve large games. In DNCFR, we use double deep neural networks to approximate these two values. DNCFR needs less memory than tabular methods because of its generalization.

probability of the opponent's hidden variable, *i.e.*, $p(x_{-i}^v|I_i) \propto \pi_{-i}^\sigma(h)$, where $x_i^v$ and $I_i$ indicate a particular $h$ (proof in Appendix D.1). Finally, the **infoset reach probability** of $I_i$ is defined as $\pi^\sigma(I_i) = \sum_{h \in I_i} \pi^\sigma(h)$. Similarly, we have $\pi_i^\sigma(I_i) = \sum_{h \in I_i} \pi_i^\sigma(h)$ and $\pi_{-i}^\sigma(I_i) = \sum_{h \in I_i} \pi_{-i}^\sigma(h)$. More details can be found in Appendix B.3.

● **Counterfactual Regret Minimization.** CFR is an iterative method for finding a Nash equilibrium for zero-sum perfect-recall IIGs (Zinkevich et al., 2007) (Algorithm 1 and Figure 2(a)). Given strategy profile $\sigma$, the **counterfactual value (CFV)** $v_i^\sigma(I_i)$ at infoset $I_i$ is defined by Eq. (1). The **action CFV** of taking action $a$ is $v_i^\sigma(a|I_i)$ and its regret is defined by Eq. (2). Then the **cumulative regret** of action $a$ after $T$ iterations is Eq. (3), where $R_i^0(a|I_i) = 0$. Define $R_i^{t,+}(a|I_i) = \max(R_i^t(a|I_i), 0)$, the **current strategy (or behavior strategy)** at $t+1$ iteration will be updated by Eq. (4). Define $s_i^t(a|I_i) = \pi_i^{\sigma^t}(I_i)\sigma_i^t(a|I_i)$ as the **additional strategy** in iteration $t$, then the **cumulative strategy** can be defined as Eq. (5), where $S^0(a|I_i) = 0$. The **average strategy** $\bar{\sigma}_i^t$ after $t$ iterations is defined by Eq. (6), which approaches a Nash equilibrium after enough iterations.

● **Monte Carlo CFR.** Lanctot et al. (2009) proposed a Monte Carlo CFR (MCCFR) to compute the unbiased estimation of counterfactual value by sampling subsets of infosets in each iteration. Although MCCFR still needs two tabular storages for saving cumulative regret and strategy as CFR does, it needs much less working memory than the standard CFR (Zinkevich et al., 2007). This is because MCCFR needs only to maintain values for those visited nodes into working memory; Define $\mathcal{Q} = \{Q_1, Q_2, ..., Q_m\}$, where $Q_j \in Z$ is a set (block) of sampled terminal histories in each iteration, such that $\mathcal{Q}_j$ spans the set $Z$. Define $q_{Q_j}$ as the probability of considering block $Q_j$, where $\sum_{j=1}^m q_{Q_j} = 1$. Define $q(z) = \sum_{j:z \in Q_j} q_{Q_j}$ as the probability of considering a particular terminal history $z$. For infoset $I_i$, an estimate of sampled counterfactual value is $\tilde{v}_i^\sigma(I_i|Q_j) = \sum_{h \in I_i, z \in Q_j, h \sqsubseteq z} \frac{1}{q(z)} \pi_{-i}^\sigma(z)\pi_i^\sigma(h,z)u_i(z)$.

**Lemma 1** *(Lanctot et al. (2009)) The sampled counterfactual value in MCCFR is the unbiased estimation of actual counterfactual value in CFR.* $E_{j \sim q_{Q_j}}[\tilde{v}_i^\sigma(I_i|Q_j)] = v_i^\sigma(I_i)$.

Define $\sigma^{rs}$ as sampled strategy profile, where $\sigma_i^{rs}$ is the sampled strategy of player $i$ and $\sigma_{-i}^{rs}$ are those for other players except for $i$. The regret of the sampled action $a \in A(I_i)$ is defined by $\tilde{r}_i^\sigma((a|I_i)|Q_j) = \sum_{z \in Q_j, ha \sqsubseteq z, h \in I_i} \pi_i^\sigma(ha, z)u_i^{rs}(z) - \sum_{z \in Q_j, h \sqsubseteq z, h \in I_i} \pi_i^\sigma(h, z)u_i^{rs}(z)$, where $u_i^{rs}(z) = \frac{u_i(z)}{\pi_i^{\sigma^{rs}}(z)}$ is a new utility weighted by $\frac{1}{\pi_i^{\sigma^{rs}}(z)}$. The sampled estimation for cumulative regret of action $a$ after $t$ iterations is $\tilde{R}_i^t((a|I_i)|Q_j) = \tilde{R}_i^{t-1}((a|I_i)|Q_j) + \tilde{r}_i^{\sigma^t}((a|I_i)|Q_j)$, where $\tilde{R}_i^0((a|I_i)|Q_j) = 0$.

# 3 DOUBLE NEURAL COUNTERFACTUAL REGRET MINIMIZATION

Double neural CFR algorithm will employ two neural networks, one for the cumulative regret $\mathcal{R}$, and the other for the average strategy $\mathcal{S}$ shown in Figure 2(b).

## 3.1 MODELING

The iterative updates of CFR algorithm maintain the regret sum $R^t(a|I_i)$ and the average strategy $\bar{\sigma}_i^t(a|I_i)$. Thus, our two neural networks are designed accordingly.

● **RegretSumNetwork(RSN):** according to Eq. (4), the current strategy $\sigma^{t+1}(a|I_i)$ is computed from the cumulative regret $R^t(a|I_i)$. We only need to track the numerator in Eq. (4) since the normalization in the denominator can be computed easily when the strategy is used. Given infoset $I_i$ and action $a$, we design a neural network $\mathcal{R}(a, I_i|\theta_\mathcal{R}^t)$ to track $R^t(a|I_i)$, where $\theta_\mathcal{R}^t$ are the network parameters.

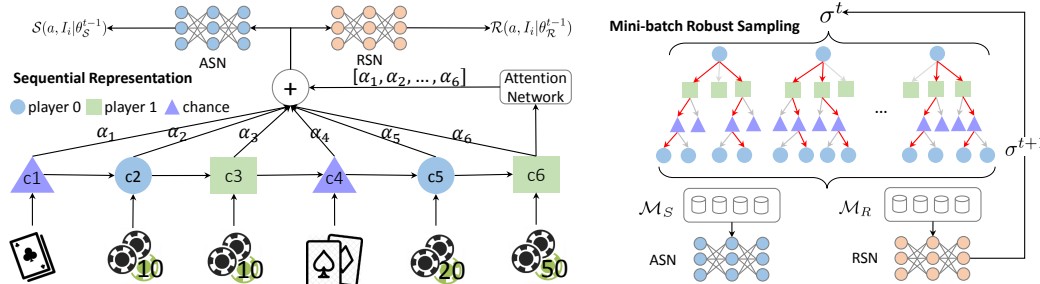

Figure 3: (a) recurrent neural network architecture with attention for extensive games. Both RSN and ASN are based on this architecture but with different parameters ($\theta_\mathcal{R}$ and $\theta_\mathcal{S}$ respectively). (b) an overview of the proposed robust sampling and mini-batch techniques. The trajectories marked by red arrows are the samples produced by robust sampling ($k=2$ here).

- **AvgStrategyNetwork(ASN):** according to Eq. (6), the approximate Nash equilibrium is the weighted average of all previous behavior strategies up to $t$ iterations, which is computed by the normalization of cumulative strategy $S^t(a|I_i)$. Similar to the cumulative regret, we employ the other deep neural network $\mathcal{S}(a|\theta_\mathcal{S}^t)$ with network parameter $\theta_\mathcal{S}^t$ to track the cumulative strategy.

## 3.2 RECURRENT NEURAL NETWORK REPRESENTATION WITH ATTENTION

In order to define our $\mathcal{R}$ and $\mathcal{S}$ networks, we need to represent the infoset in extensive-form games. In such games, players take actions in an alternating fashion and each player makes a decision according to the observed history. Because the action sequences vary in length, we model them with recurrent neural networks and each action in the sequence corresponds to a cell in the RNN. This architecture is different from the one in DeepStack (Moravcik et al., 2017), which used a fully connected deep neural network to estimate counterfactual value. Figure 3(a) provides an illustration of the proposed deep sequential neural network representation for infosets. Besides the vanilla RNN, there are several variants of more expressive RNNs, such as the GRU (Cho et al., 2014) and LSTM (Hochreiter & Schmidhuber, 1997). In our later experiments, we will compare these different neural architectures as well as a fully connected network representation.

Furthermore, different position in the sequence may contribute differently to the decision making, we add an attention mechanism (Desimone & Duncan, 1995; Cho et al., 2015) to the RNN architecture to enhance the representation. For example, the player may need to take a more aggressive strategy after beneficial public cards are revealed in a poker game. Thus the information after the public cards are revealed may be more important. In practice, we find that the attention mechanism can help DNCFR obtain a better convergence rate. See Appendix E for more details on the architectures.

## 3.3 OPTIMIZATION METHOD

The parameters in the two neural networks are optimized via stochastic gradient descent in a stage-wise fashion interleaving with CFR iterations.

### 3.3.1 OPTIMIZING CURRENT STRATEGY

We use $\mathcal{M}_R^t = \{(I_i, \tilde{r}_i^{\sigma^t}((a|I_i)|Q_j))|\text{for all sampled } I_i\}$ to store the sampled $I_i$ and the corresponding regret $\tilde{r}_i^{\sigma^t}((a|I_i)|Q_j))$ for all players in $t$-th iteration, where $Q_j$ is the sampled block (shown in Figure 2(b)). These samples are produced by our proposed robust sampling and mini-batch MCCFR methods, which will be discussed in Section 4. According to Eq. (3), we optimize the cumulative regret neural network $\mathcal{R}(a, I_i|\theta_\mathcal{R}^{t+1})$ using the following loss function

$$\mathcal{L}(\mathcal{R}) = \sum_{\substack{i \in N, \\ I_i \in \mathcal{I}_i, \\ a \in A(I_i)}} \begin{cases} \left( \mathcal{R}(\cdot|\theta_\mathcal{R}^t) + \tilde{r}_i^{\sigma^t}(\cdot|Q_j) - \mathcal{R}(\cdot|\theta_\mathcal{R}^{t+1}) \right)^2 & \text{if } I_i \text{ in } \mathcal{M}_R^t \\ \left( \mathcal{R}(\cdot|\theta_\mathcal{R}^t) + 0 - \mathcal{R}(\cdot|\theta_\mathcal{R}^{t+1}) \right)^2 & \text{otherwise,} \end{cases} \tag{7}$$

where $\mathcal{R}((a|I_i)|\theta_\mathcal{R}^t)$ refers to $\mathcal{R}(\cdot|\theta_\mathcal{R}^t)$, $\tilde{r}_i^{\sigma^t}((a|I_i)|Q_j)$ refers to $\tilde{r}_i^{\sigma^t}(\cdot|Q_j)$, $\theta_\mathcal{R}^t$ refers to the old parameters and $\theta_\mathcal{R}^{t+1}$ is the new parameters we need to optimize. Note that, Eq. (7) is minimized based on the samples of all the players rather than a particular player $i$. In standard MCCFR, if the infoset is not sampled, the corresponding regret is set to 0, which leads to unbiased estimation according to Lemma 1. The design of the loss function in Eq. (7) follows the same intuition. Techniques in Schmid et al. (2018) can be used to reduce the variance.

**Sampling unobserved infosets?** Theoretically, in order to optimize Eq. (7), we need to collect both observed and unobserved infosets. This approach requires us to design a suitable sampling method to select additional training samples from large numbers of unobserved infosets, which will need a lot of memory and computation. Clearly, this is intractable on large games, such as HUNL. In practice, we find that minimizing loss only based on the observed samples can help us achieve a converged strategy.

**Learning without forgetting?** Another concern is that, only a small proportion of infosets are sampled due to mini-batch training, which may result in the neural networks forgetting values for those unobserved infosets. To address this challenge, we will use the neural network parameters from the previous iteration as the initialization, which gives us an online learning/adaptation flavor to the updates. Experimentally, on large games, due to the generalization ability of the neural networks, even a small proportion of infosets are used to update the neural networks, our double neural approach can still converge to an approximate Nash equilibrium. See Appendix F for more details on implementation.

**Scaling regret for stable training?** According to Theorem 6 in Burch (2017), the cumulative regret $R_i^t(a|I_i) \leq \Delta\sqrt{|A|T}$, where $|A| = \max_{I_i \in \mathcal{I}} |A(I_i)|$ and $\Delta = \max_{I_i, a, t} |R^t(a|I_i) - R^{t-1}(a|I_i)|$. It indicates that $R_i^t(a|I_i)$ will become increasingly large. In practice, we scale the cumulative regret by a factor of $\sqrt{t}$ to make its range more stable. For example, define $\hat{R}_i^t(a|I_i) = R_i^t(a|I_i)/\sqrt{t}$, we can update the cumulative regret Eq. (3) by $\hat{R}_i^t(a|I_i) = (\sqrt{t-1}\hat{R}_i^{t-1}(a|I_i) + r_i^{\sigma^t}(a|I_i))/\sqrt{t}$, where $\hat{R}_i^0(a|I_i) = 0$.

### 3.3.2 Optimizing Average Strategy

The other memory $\mathcal{M}_S^t = \{(I_i, s_i^t(a|I_i)|\text{for all sampled } I_i\}$ will store the sampled $I_i$ and the weighted additional behavior strategy $s_i^t(a|I_i)$ in $t$-th iteration. Similarly, the loss function $\mathcal{L}(\mathcal{S})$ of ASN is defined by:

$$\mathcal{L}(\mathcal{S}) = \sum_{\substack{i \in N, \\ I_i \in \mathcal{I}_i, \\ a \in A(I_i)}} \begin{cases} \left(\mathcal{S}(\cdot|\theta_\mathcal{S}^t) + s_i^t(a|I_i) - \mathcal{S}(\cdot|\theta_\mathcal{S}^{t+1})\right)^2 & \text{if } I_i \text{ in } \mathcal{M}_S^t \\ \left(\mathcal{S}(\cdot|\theta_\mathcal{S}^t) + 0 - \mathcal{S}(\cdot|\theta_\mathcal{S}^{t+1})\right)^2 & \text{otherwise.} \end{cases} \tag{8}$$

where $\mathcal{S}(\cdot|\theta_\mathcal{S}^t)$ refers to $\mathcal{S}(a, I_i|\theta_\mathcal{S}^t)$, $\theta_\mathcal{S}^t$ refers to the old parameters and $\theta_\mathcal{S}^{t+1}$ is the new parameters we need to optimize. According to Algorithm 1, cumulative regret is used to generate behavior strategy in the next iteration while cumulative strategy is the summation of the weighted behavior strategy. In theory, if we have all the $\mathcal{M}_S^t$ in each iteration, we can achieve the final average strategy directly. Based on this concept, we don't need to optimize the average strategy network (ASN) $\mathcal{S}(\cdot|\theta_\mathcal{S}^t)$ in each iteration. However, saving all such values into a huge memory is very expensive on large games. A compromise is that we can save such values within multiple iterations into a memory, when this memory is large enough, the incremental value within multiple iterations can be learned by optimizing Eq. (8).

**Minimum squared loss versus maximum likelihood?** The average strategy is a distribution over actions, which implies that we can use maximum likelihood method to directly optimize this average strategy. The maximum likelihood method should base on the whole samples up to $t$-th iteration rather than only the additional samples, so that this method is very memory-expensive. To address this limitation, we can use uniform reservoir sampling method (Osborne et al., 2014) to obtain the unbiased estimation of each strategy. In practice, we find this maximum likelihood method has high variance and cannot approach a less exploitable Nash equilibrium. Experimentally, optimization by minimizing squared loss helps us obtain a fast convergent average strategy profile and uses much less memory than maximum likelihood method.

### 3.4 Continual Improvement

When solving large IIGs, prior methods such as Libratus (Brown & Sandholm, 2017) and Deep-Stack (Moravcik et al., 2017) are based on the abstracted HUNL which has a manageable number of infosets. The abstraction techniques are usually based on domain knowledge, such as clustering similar hand-strength cards into the same buckets or only taking discrete actions (*e.g.*, fold, call, one-pot raise and all in). DNCFR is not limited by the specified abstracted cards or actions. For example, we can use the continuous variable to represent bet money rather than encode it by discrete action. In practice, DNCFR can clone an existing tabular representation or neural representation and then continually improve the strategy from the initialized point. More specifically, for infoset $I_i$ and action $a$, define $R_i'(a|I_i)$ as the cumulative regret . We can use behavior cloning technique to learn the cumulative regret by optimizing $\theta_\mathcal{R}^* \leftarrow \text{argmin}_{\theta_\mathcal{R}} \sum_{I_i \in \mathcal{I}_i} \left(\mathcal{R}(\cdot|\theta_\mathcal{R}) - R'(\cdot|I_i)\right)^2$. Similarly, the cumulative strategy can be cloned in the

same way. Based on the learned parameters, we can warm start DNCFR and continually improve beyond the tabular strategy profile.

### 3.5 OVERALL ALGORITHM

Algorithm 2 provides a summary of the proposed double neural counterfactual regret minimization approach. In the first iteration, if the system warm starts from tabular-based methods, the techniques in Section 3.4 will be used to clone the cumulative regrets and strategies. If there is no warm start initialization, we can start our algorithm by randomly initializing the parameters in RSN and ASN. Then sampling methods will return the sampled infosets and values, which are saved in memories $\mathcal{M}_{\mathcal{R}}^t$ and $\mathcal{M}_{\mathcal{S}}^t$ respectively. These samples will be used by the NeuralAgent algorithm from Algorithm 3 to optimize RSN and ASN. Further details for the sampling methods will be discussed in the next section. Due to space limitation, we present NeuralAgent fitting algorithm in Appendix F.

---

**Algorithm 2:** DNCFR Algorithm

**Function** Agent $(T, b)$**:**
   **For** $t=1$ *to* $T$ **do**
      **if** $t=1$ ***and** using warm starting* **then**
         | Initialize $\theta_{\mathcal{R}}^t$ and $\theta_{\mathcal{S}}^t$ from a checkpoint
         |    $t \leftarrow t+1$
      **else**
         | Initialize $\theta_{\mathcal{R}}^t$ and $\theta_{\mathcal{S}}^t$ randomly.
      $\mathcal{M}_{\mathcal{R}}^t, \mathcal{M}_{\mathcal{S}}^t \leftarrow$ sampling methods.
      Sum aggregate value in $\mathcal{M}_R$ by infoset.
      Remove duplicated records in $\mathcal{M}_S$.
      $\theta_{\mathcal{R}}^t \leftarrow$ NeuralAgent$(\mathcal{R}(\cdot|\theta_{\mathcal{R}}^{t-1}), \mathcal{M}_R^t, \theta_{\mathcal{R}}^{t-1}, \beta_{\mathcal{R}}^*)$
      $\theta_{\mathcal{S}}^t \leftarrow$ NeuralAgent$(\mathcal{S}(\cdot|\theta_{\mathcal{S}}^{t-1}), \mathcal{M}_S^t, \theta_{\mathcal{S}}^{t-1}, \beta_{\mathcal{S}}^*)$
   **return** $\theta_{\mathcal{R}}^t, \theta_{\mathcal{S}}^t$

---

## 4 EFFICIENT TRAINING

In this section, we will propose two techniques to improve the efficiency of the double neural method. These techniques can also be used separately in other CFR variants.

### 4.1 ROBUST SAMPLING TECHNIQUE

In this section, we introduce a robust sampling method (RS), which is a general version of both external sampling and outcome sampling (Lanctot et al., 2009). RS samples $k$ actions in one player's infosets and samples one action in the another player's infosets. Specifically, in the robust sampling method, the sampled profile is defined by $\sigma^{rs(k)} = (\sigma_i^{rs(k)}, \sigma_{-i})$, where player $i$ will randomly select $k$ actions according to sampled strategy $\sigma_i^{rs(k)}(I_i)$ at $I_i$ and other players randomly select one action according to $\sigma_{-i}$.

We design an efficient sampling policy for robust sampling as follows and discuss the relationship among robust sampling, external sampling and outcome sampling in Appendix D.2. If $k = max_{I_i \in \mathcal{I}}|A(I_i)|$ and for each action $\sigma_i^{rs(k)}(a|I_i) = 1$, then robust sampling is identical with external sampling. If $k=1$, $\sigma_i^{rs(k)} = \sigma_i$ and $q(z) \geq \delta > 0$ ($\delta$ is a small positive number), then robust sampling is identical with outcome sampling.

Specifically, if player $i$ randomly selects $\min(k, |A(I_i)|)$ actions according to discrete uniform distribution $unif(0, |A(I_i)|)$ at $I_i$, *i.e.*, $\sigma_i^{rs(k)}(a|I_i) = \frac{\min(k,|A(I_i)|)}{|A(I_i)|}$, then $\pi_i^{\sigma^{rs(k)}}(I_i) = \prod_{h \in I_i, h' \sqsubseteq h, h'a \sqsubseteq h, h' \in I_i'} \frac{min(k,|A(I_i')|)}{|A(I_i')|}$ and the weighted utility $u_i^{rs(k)}(z)$ will be a constant number in each iteration. In many settings, when $k=1$, we find such robust sampling schema converges more efficient than outcome sampling. In contrast, our robust sampling achieves comparable convergence with external sampling but using less working memory when specifying a suitable $k$. It's reasonable because our schema only samples $k$ rather than all actions in player $i's$ infosets, the sampled game tree is smaller than the one by external sampling. In the experiment, we will compare these sampling policies in our ablation studies.

### 4.2 MINI-BATCH TECHNIQUE

Traditional MCCFR only samples one block in an iteration and provides an unbiased estimation of origin CFV. In this paper, we present a mini-batch Monte Carlo technique and randomly sample $b$ blocks in one iteration. Let $Q^j$ denote a block of terminals sampled according to the scheme in Section 4.1, then **mini-batch CFV** with mini-batch size $b$ will be $\tilde{v}_i^\sigma(I_i|b) = \sum_{j=1}^b \sum_{h \in I_i, h \sqsubseteq z, z \in Q^j} \pi_{-i}^\sigma(z)\pi_i^\sigma(h,z)u_i(z)/(bq(z))$.

**Theorem 1** $E_{Q^j \sim mini\text{-}batch}[\tilde{v}_i^\sigma(I_i|b)] = v_i^\sigma(I_i)$.

We prove that $\tilde{v}_i^\sigma(I_i|b)$ is an unbiased estimation of CFV in Appendix D.3. Following the similar ideas of CFR and CFR+, if we replace the regret matching by regret matching plus (Tammelin, 2014), we obtain a mini-batch MCCFR+ algorithm. Our mini-batch technique empirically can sample $b$ blocks in parallel and converges faster than original MCCFR when performing on multi-core machines.

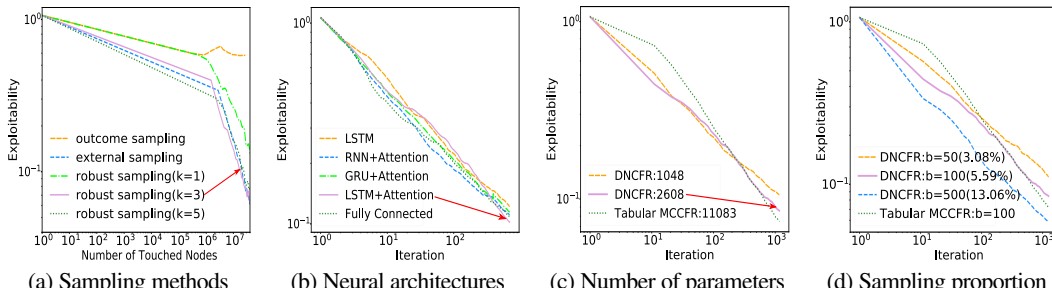

(a) Sampling methods     (b) Neural architectures     (c) Number of parameters     (d) Sampling proportion

Figure 4: Log-log performance on Leduc(5). (a) different sampling methods, $k$ refers to the number of sampling action for the proposed robust sampling method in each infoset. (b) neural architectures. (c) number of parameters. (d) proportion of observed infosets. Higher proportion indicates more working memory.

## 5 EXPERIMENT

To understand the contributions of various components in DNCFR algorithm, we will first conduct a set of ablation studies. Then we will compare DNCFR with tabular CFR and deep reinforcement learning method such as NFSP, which is a prior leading function approximation method in IIGs. At last, we conduct experiments on heads-up no-limit Texas Hold'em (HUNL) to show the scalability of DNCFR algorithm. The games and key information used in our experiment are listed in Table 1.

Table 1: Summary. #infoset is the number of infosets. #state is the number of states. %observed is the ratio of observed infosets in each iteration. #emd and #param are the embedding size and the number of parameters in DNCFR.

| setting | #infoset | #state | %observed | #emd | #param | action abstraction | card abstraction |
|---|---|---|---|---|---|---|---|
| Leduc(5) | $1 \times 10^4$ | $6 \times 10^5$ | 5.59% | 16 | 2608 | No | No |
| Leduc(10) | $3 \times 10^5$ | $2 \times 10^6$ | 2.39% | 32 | 7424 | No | No |
| Leduc(15) | $3 \times 10^6$ | $2 \times 10^7$ | 0.53% | 64 | 23360 | No | No |
| HUNL(1) | $2 \times 10^8$ | $3 \times 10^{11}$ | 0.01% | 64 | 19200 | Yes | No |
| HUNL(2) | $8 \times 10^{10}$ | $1 \times 10^{14}$ | 0.001% | 512 | 1070592 | Yes | No |
| ABS-CFR | $2 \times 10^{10}$ | $2 \times 10^{13}$ | – | – | – | Yes | Yes |
| HUNL(full) | $1 \times 10^{161}$ | $1 \times 10^{164}$ | – | – | – | Yes | No |

### 5.1 SETTINGS AND METRIC

We perform the ablation studies on Leduc Hold'em poker, which is a commonly used poker game in research community (Heinrich & Silver, 2016; Schmid et al., 2018; Steinberger, 2019; Lockhart et al., 2019). In our experiments, we test DNCFR on three Leduc Hold'em instances with stack size 5, 10, and 15, which are denoted by Leduc(5), Leduc(10), and Leduc(15) respectively.

To test DNCFR's scalability, we develop a neural agent to solve HUNL, which contains about $10^{161}$ infosets (Johanson, 2013) and has served for decades as challenging benchmark and milestones of solving IIGs. The rules for such games are given in Appendix A.

The experiments are evaluated by exploitability, which was used as a standard win rate measure in many key articles (Zinkevich et al., 2007; Lanctot et al., 2009; Michael Bowling, 2015; Brown et al., 2018). The units of exploitability in our paper is chips per game. It denotes how many chips one player wins on average per hand of poker. The method with a lower exploitability is better. The exploitability of Nash equilibrium is zero. In extremely large game, which is intractable to compute exploitability, we use head-to-head performance to measure different agents.

For reproducibility, we present the implementation details of the neural agent in Algorithm 2, Algorithm 3, Algorithm 4. Appendix F.4 provides the parameters used in our experiments. Solving HUNL is a challenging task. Although there are published papers (Moravcik et al., 2017; Brown & Sandholm, 2017), it lacks of available open source codes for such solvers. The development of HUNL solver not only needs tedious work, but also is difficult to verify the correctness of the implementation, because of its well known high variance and extremely large game size. In Appendix G, we provide several approaches to validate the correctness of our implementation for HUNL.

### 5.2 ABLATION STUDIES

We first conduct a set of ablation studies related to the mini-batch training, robust sampling, the choice of neural architecture on Leduc Hold'em.

- **Is mini-batch sampling helpful?** we present the convergence curves of the proposed robust sampling method with $k = \max(|A(I_i)|)$ under different mini-batch sizes in Figure 8(a) at Appendix C. The experimental results show that larger batch sizes generally lead to better strategy profiles.

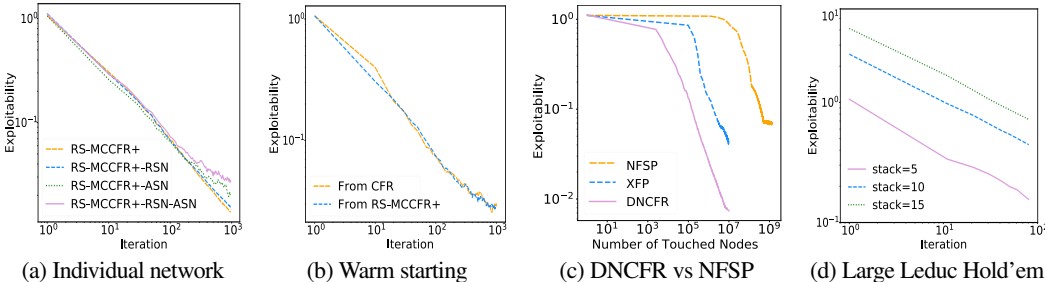

(a) Individual network  (b) Warm starting  (c) DNCFR vs NFSP  (d) Large Leduc Hold'em

Figure 5: Log-log performance. (a) Individual effect of RSN and ASN. RS-MCCFR+ refers to the tabular mini-batch MCCFR+ method with the proposed robust sampling. RS-MCCFR+-RSN only uses one neural network RSN to learn cumulative regret while uses a table to save cumulative strategy. RS-MCCFR+-ASN only use one neural network ASN. RS-MCCFR+-RSN-ASN refers to DNCFR with both RSN and ASN. (b) Warm start from tabular CFR and RS-MCCFR+. (c) DNCFR vs XFP vs NFSP. (d) Large Leduc(10) and Leduc(15).

- **Is robust sampling helpful?** Figure 4 (a) presents convergence curves for outcome sampling, external sampling($k = \max(|A(I_i)|)$) and the proposed robust sampling method under the different number of sampled actions. The outcome sampling cannot converge to a low exploitability( smaller than 0.1 after 1000 iterations). The proposed robust sampling algorithm with $k = 1$, which only samples one trajectory like the outcome sampling, can achieve a better strategy profile after the same number of iterations. With an increasing $k$, the robust sampling method achieves an even better convergence rate. Experiment results show $k = 3$ and $5$ have a similar trend with $k = \max(|A(I_i)|)$, which demonstrates that the proposed robust sampling achieves similar performance but requires less memory than the external sampling. We choose $k = 3$ for the later experiments in Leduc Hold'em.
- **Is attention in the neural architecture helpful?** Figure 4(b) shows that all the neural architectures achieved similar results while LSTM with attention achieved slightly better performance with a large number of iterations. We select LSTM plus attention as the default architectures in the later experiments.
- **Do the neural networks just memorize but not generalize?** One indication that the neural networks are generalizing is that they use much fewer parameters than their tabular counterparts. We experimented with LSTM plus attention networks, and embedding size of 8 and 16 respectively. These architectures contain 1048 and 2608 parameters respectively. Both of them are much less than the tabular memory (more than 11083 here) and can lead to a converging strategy profile as shown in Figure 4(c). We select embedding size 16 as the default parameters. In the later experiments, we will show the similar conclusion on HUNL.
- **Do the neural networks generalize to unseen infosets?** To investigate the generalization ability, we perform the DNCFR with small mini-batch sizes (b=50, 100, 500), where only $3.08\%$, $5.59\%$, and $13.06\%$ infosets are observed in each iteration. In all these settings, DNCFR can still converge and arrive at exploitability less than 0.1 within only 1000 iterations as shown in Figure 4(d). In the later experiments, we set b=100 as the default mini-batch size. We learn new parameters based on the old parameters and a subset of observed samples. All infosets share the same parameters, so that the neural network can estimate the values for unseen infosets. Note that, the number of parameters is orders of magnitude less than the number of infosets in many settings, which indicates the generalization of our method. Furthermore, Figure 4(d) shows that DNCFRs are slightly better than tabular MCCFR, we think it's because of the generalization to unseen infosets.
- **What is the individual effect of RSN and ASN?** Figure 5(a) presents ablation study of the effects of RSN and ASN network respectively. Specifically, the method RSN denotes that we only employ RSN to learn the cumulative regret while the cumulative strategy is stored in a tabular memory. Similarly, the method ASN only employ ASN to learn the cumulative strategy. Both these single neural methods perform only slightly better than the DNCFR.
- **How well does continual improvement work?** As shown in Figure 5(b), warm starting from either full-width based or sampling based CFR can lead to continual improvements. Specifically, the first 10 iterations are learned by tabular based CFR and RS-MCCFR+. After the behavior cloning in Section 3.4, the remaining iterations are continually improved by DNCFR.
- **How well does DNCFR on larger games?** We test DNCFR on large Leduc(10) and Leduc(15), which contains millions of infosets. Even though only a small proportion of nodes are sampled in each iteration, Figure 5(d) shows that DNCFR can still converge on these large games.

## 5.3 COMPARISON AND SPACE-TIME TRADE-OFF

**How does DNCFR compare to the tabular counterpart, XFP, and NFSP?** NFSP is the prior leading function approximation method for solving IIG, which is based on reinforcement learning and fictitious self-play techniques. In the experiment, NFSP requires two memories to store $2 \times 10^5$ state-action pair

samples and $2 \times 10^6$ samples for supervised learning respectively. The memory sizes are larger than the number of infosets. Figure 5(c) demonstrates that NFSP obtains a 0.06-Nash equilibrium after touching $10^9$ infosets. The XFP obtains the same exploitability when touching about $10^7$ nodes. However, this method is the precursor of NFSP and updated by a tabular based full-width fictitious play. Our DNCFR achieves the same performance by touching no more than $10^6$ nodes, which are much fewer than both NFSP and XFP. The experiment shows that DNCFR converges significantly better than the reinforcement learning counterpart.

**Space and time trade-off.** In this experiment, we investigate the time and space needed for DNCFR to achieve certain exploitability relative to tabular CFR algorithm. We compare their runtime and memory in Figure 6. It's clear that the number of infosets is much more than the number of parameters used in DNCFR. For example, on Leduc(15), tabular CFR needs 128 times more memory than DNCFR. In the figure, we use the ratio between the runtime of DNCFR and CFR as horizontal axis, and the sampling(observed) infosets ratios of DNCFR and full-width tabular CFR as vertical axis. Note that, the larger the sampling ratio, the more memory will be needed to save the sampled values.

Clearly, there is a trade-off between the relative runtime and relative memory in DNCFR: the longer the relative runtime, the less the relative memory needed for DNCFR. It is reasonable to expect that a useful method should lead to "fair" trade between space and time. That is onefold increase in relative runtime should lead onefold decreases in relative memory (the dashed line in Figure 6, slope -1). Interestingly, DNCFR achieves a much better trade-off between relative runtime and memory: for onefold increases in relative runtime, DNCFR may lead to fivefold decreases in relative memory consumption (red line, slope -5). We believe this is due to the generalization ability of the learned neural networks in DNCFR.

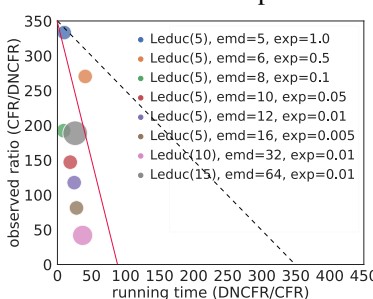

Figure 6: time space trade-off.

To present the time space trade off under a range of exploitability, we set the fixed exploitability as 1.0, 0.5, 0.1, 0.05, 0.01 and 0.005 and perform both neural and tabular CFR on Leduc Hold'em. Figure 6 presents DNCFR achieves a much better time and space trade-off. We believe the research on neural CFR is important for future work and the running time is not the key limitation of our DNCFR. Some recent works (Schmid et al., 2018; Davis et al., 2019) provide strong variance reduction techniques for MCCFR and suggest promising direction for DNCFR. In the future, we will combine DNCFR with the latest acceleration techniques and use multiple processes or distributed computation to make it more efficient.

### 5.4 Heads-up No-Limit Texas Hold'em

To test the scalability of the DNCFR on extremely large game, we develop a neural agent to solve HUNL. However, it's a challenging task to directly solve HUNL even with abstraction technique. For example, ABS-CFR uses k-means to cluster similar cards into thousands of clusters. Although it's a rough abstraction of original HUNL, such agent contains about $2 \times 10^{10}$ infosets and needs 80GB memory to store its strategies. The working memory for training ABS-CFR is even larger (more than about 200GB), because it needs to store cumulative regrets and other essential variables, such as the abstracted mapping. To make it tractable for solving HUNL via deep learning, we assemble the ideas from both DeepStack (Moravcik et al., 2017) and Libratus (Brown & Sandholm, 2017). Firstly, we train flop and turn networks like DeepStack and use these networks to predict counterfactual value when given two players' ranges and the pot size. Specifically, the flop network estimates values after dealing the first three public cards and the turn network estimates values after dealing the fourth public card. After that, we train blueprint strategies like Libratus. In contrast, the blueprint strategies in our settings are learned by DNCFR. Because we have value networks to estimate counterfactual values, there is no need for us to arrive at terminal nodes at the river.

To demonstrate the convergence of DNCFR, firstly, we test it on HUNL(1). Such game has no limited number of actions, contains four actions in each infoset, and ends with the terminals where the first three public cards are dealt. HUNL(1) contains more than $2 \times 10^8$ infosets and $3 \times 10^{11}$ states. It's tractable to compute its exploitability within the limited time. We believe this game is suitable to evaluate the scalability and generalization of DNCFR. Figure 7(a) provides the convergence of DNCFR on different embedding size: emd=8, 16, 32, 64, 128. The smallest neural network only contains 608 parameters while the largest one contains 71168 parameters. It's reasonable to expect that a larger neural network typically achieves better performance because more parameters typically help neural networks represent more complicated patterns and structures. Figure 7(b) presents the performance of using the different number of stochastic gradient descent (SGD) updates to train neural network on each MCCFR iteration. The results show that the number of SGD updates on each iteration affects the asymptotic exploitability of DNCFR. It's

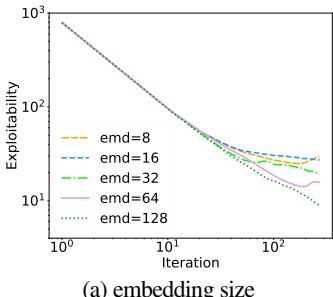 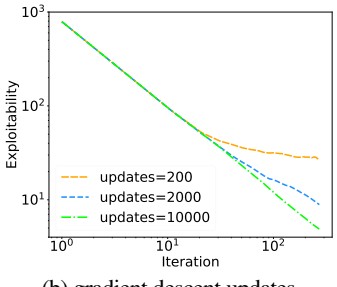

| Match-up | Win(chips/h) |
|---|---|
| ABS-CFR | $9.8\pm4.1$ |
| Tabular Agent | $0.7\pm2.2$ |

(a) embedding size  (b) gradient descent updates  (c) head-to-head win rate.

Figure 7: Performance of DNCFR on heads-up no-limit Texas Hold'em. (a) Log-log performance of DNCFR on HUNL(1) under different embedding size. (b) Log-Log performance of DNCFR on HUNL(1) under different numbers of gradient descent updates on each iteration. (c) DNCFR beats ABS-CFR by $9.8\pm4.1$ chips per hand and achieves similar performance with its tabular version but using much less memory.

reasonable because the neural network achieves small loss as the number of gradient descent updates is increasing.

Finally, we measure the head-to-head performance of our neural agent against its tabular version and ABS-CFR on HUNL. ABS-CFR is a strong HUNL agent, which is the advanced version of the third-place agent in ACPC 2018. Although ABS-CFR used both card and action abstraction techniques, it still needs 80GB memory to store its strategies. More details about ABS-CFR are provided in Appendix G.1. Although abstraction pathologies are well known in extensive games (Waugh et al., 2009), typically, finer grained abstraction leads to better strategy in many settings. Following this idea, we use DNCFR to learn blueprint strategies on HUNL(2), which is similar to HUNL(1) but contains eight actions in each infoset. HUNL(2) contains $8\times10^{10}$ infosets. Such large game size makes it intractable to perform subgame solving (Burch et al., 2014) in real-time. For the next rounds, we use continual resolving techniques to compute strategy in real-time. The action size in the look-ahead tree is similar to Table S3 in Moravcik et al. (2017). The tabular agent is similar to our neural agent except for using tabular CFR to learn blueprint strategies. When variance reduction techniques (Burch et al., 2018) are applied [3], Figure 7(c) shows that our neural agent beats ABS-CFR by $9.8\pm4.1$ chips per game and obtains similar performance ($0.7\pm2.2$ chips per game) with its tabular agent. In contrast, our neural only needs to store 1070592 parameters, which uses much less memory than both tabular agent and ABS-CFR.

## 6 RELATED WORKS AND DISCUSSION

Solving IIGs via function approximation methods is an important and challenging problem. Neural Fictitious Self-Play (NFSP) (Heinrich & Silver, 2016) is a function approximation method based on deep reinforcement learning, which is a prior leading method to solve IIG. However, fictitious play empirically converges slower than CFR-based approaches in many settings. Recently, Lockhart et al. (2019) propose a new framework to directly optimize the final policy against worst-case opponents. However, the authors consider only small games. Regression CFR (RCFR) (Waugh et al., 2015) is a function approximation method based on CFR. However, RCFR needs to traverse the full game tree. Such traversal is intractable in large games. In addition, RCFR uses hand-crafted features and regression tree to estimate cumulative regret rather than learning features from data. Deep learning empirically performs better than regression tree in many areas, such as the Transformer and BERT in natural language models (Ashish Vaswani, 2017; Jacob Devlin, 2018).

In the past year, concurrent works deep CFR (DCFR) (Brown et al., 2018) and single deep CFR (SD-CFR) (Steinberger, 2019) have been proposed to address this problem via deep learning. DCFR, SDCFR, RCFR and our DNCFR are based on the framework of counterfactual regret minimization. However, there are many differences in several important aspects, which are listed as follows. (1) We represent the extensive-form game by recurrent neural network. The proposed LSTM with attention performs better than fully connected network (see details in Section 3.2). (2) DNCFR updates the cumulative regret only based on the additionally collected samples in current iteration rather than using the samples in a big reservoir (see details in Section 3.3.1). (3) It's important to use squared-loss for the average strategies rather than log loss. Because the log loss is based on the big reservoir samples up to $T$-th iteration, it is very memory-expensive (see details in Section 3.3.2). (4) Another important aspect to make deep learning model work is that we divide regret by $\sqrt{T}$ and renormalize the regret, because the cumulative regret can grow unboundedly

---

[3] It's well known that head-to-head evaluation of HUNL is challenging because of its high variance. AIVAT is the state-of-the-art technique to reduce evaluation variance on poker evaluation.

(see details in Section 3.3.1). (5) Also, DNCFR collects data by an efficiently unbiased mini-batch robust sampling method, which may be of independent interests to the IIG communities (see details in Section 4).

There are also big differences in the experimental evaluations. In our method, we conduct a set of ablation studies in various settings. We believe that our ablation studies are informative and could have a significant impact on these kinds of algorithms. Also, we evaluate DNCFR on extremely large games while RCFR and SDCFR are only evaluated on small toy games.

## 7 CONCLUSIONS

We proposed a novel double neural counterfactual regret minimization approach to solve large IIGs by combining many novel techniques, such as recurrent neural representation, attention, robust sampling, and mini-batch MCCFR. We conduct a set of ablation studies and the results show that these techniques may be of independent interests. This is a successful application of applying deep learning into large IIG. We believe DNCFR and other related neural methods open up a promising direction for future work.

## 8 ACKNOWLEDGMENTS

We would like to thank Marc Lanctot, Neil Burch, Noam Brown, Tuomas Sandholm, Richard Gibson, Matej Moravcik, Martin Schmid, Viliam Lisy, Dustin Morrill, Nolan Bard, Trevor Davis, Kevin Waugh, Michael Johanson and Michael Bowling for their helps in the work of developing large-scale heads-up no-limit Texas Hold'em. We appreciate the help of Zhibang Ge and Tao Jiang for their early exploration on this difficult project. We thank Lin Wang, Jun Zhou, Yongchao Liu, Yewei Huang, Junping Zhao, Jun Jiang, Jincan Kong, Shaohua Du, Yao Zhang and Changhua He for supporting computation resource. We thank Chuanxiao Zhou, Shuai Xiao, Xiang Li and Yaxi Zhu et.al. for their suggestions and professional testing. At last, we would like to thank the anonymous reviewers for their valuable feedback.

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

## A  GAME RULES

### A.1  ONE-CARD POKER

One-Card Poker is a two-players IIG of poker described by Gordon (2005). The game rules are defined as follows. Each player is dealt one card from a deck of $X$ cards. The first player can pass or bet, If the first player bet, the second player can call or fold. If the first player pass, the second player can pass or bet. If the second player bet, the first player can fold or call. The game ends with two pass, call, fold. The fold player will lose 1 chip. If the game ends with two passes, the player with higher card wins 1 chip, If the game ends with call, the player with higher card wins 2 chips.

### A.2  LEDUC HOLD'EM

Leduc Hold'em a two-players IIG of poker, which was first introduced in Southey et al. (2012). In Leduc Hold'em, there is a deck of 6 cards comprising two suits of three ranks. The cards are often denoted by king, queen, and jack. In Leduc Hold'em, the player may wager any amount of chips up to a maximum of that player's remaining stack. There is also no limit on the number of raises or bets in each betting round. There are two rounds. In the first betting round, each player is dealt one card from a deck of 6 cards. In the second betting round, a community (or public) card is revealed from a deck of the remaining 4 cards. In this paper, we use Leduc(x) refer to the Leduc Hold'em with stack size is $x$.

### A.3  HEADS-UP NO-LIMIT TEXAS HOLD'EM

Heads-Up No-Limit Texas hold'em (HUNL) has at most four betting rounds if neither of two players fold during playing. The four betting rounds are preflop, flop, turn, river respectively. The rules are defined as follows. In Annual Computer Poker Competition (ACPC), two players each have 20000 chips initially. One player at the position of small blind, firstly puts 50 chips in the pot, while the other player at the big blind then puts 100 chips in the pot. After that, the first round of betting is followed. If the preflop betting round ends without a player folding, then three public cards are revealed face-up on the table and the flop betting round occurs. After this round, one more public card is dealt (called the turn) and the third round of betting takes place, followed by a fifth public card (called river) and a final round of betting begins. In no-limit poker player can take fold, call and bet actions and bet number is from one big blind to a number of chips a player has left in his stack.

## B DEFINITION OF EXTENSIVE-FORM GAMES

### B.1 DETAILED DEFINITIONS AND NOTATIONS

We define the components of an extensive-form game following Osborne & Rubinstein (1994) (page $200 \sim 201$). A finite set $N = \{0, 1, ..., n-1\}$ of **players**. Define $x_i^v$ as the **hidden variable** of player $i$ in IIG, *e.g.*, in poker game $x_i^v$ refers to the private cards of player $i$. $H$ refers to a finite set of histories. Each member $h = (x_i^v)_{i=0,1,...,n-1}(a_l)_{l=0,...,L-1} = x_0^v x_1^v ... x_{n-1}^v a_0 a_1 ... a_{L-1}$ of $H$ denotes a possible **history** (or state), which consists of each player's hidden variable and $L$ actions taken by players including chance. For player $i$, $h$ also can be denoted as $x_i^v x_{-i}^v a_0 a_1 ... a_{L-1}$, where $x_{-i}^v$ refers to the opponent's hidden variables. The empty sequence $\emptyset$ is a member of $H$. $h_j \sqsubseteq h$ denotes $h_j$ is a prefix of $h$, where $h_j = (x_i^v)_{i=0,1,...,n-1}(a_l)_{l=1,...,L'-1}$ and $0 < L' < L$. $Z \subseteq H$ denotes the terminal histories and any member $z \in Z$ is not a prefix of any other sequences. $A(h) = \{a : ha \in H\}$ is the set of available actions after non-terminal history $h \in H \setminus Z$. A **player function** $P$ assigns a member of $N \cup \{c\}$ to each non-terminal history, where $c$ denotes the chance player id, which usually is -1. $P(h)$ is the player who takes an action after history $h$.

$\mathcal{I}_i$ of a history $\{h \in H : P(h) = i\}$ is an **information partition** of player $i$. A set $I_i \in \mathcal{I}_i$ is an **information set (infoset)** of player $i$ and $I_i(h)$ refers to infoset $I_i$ at state $h$. Generally, $I_i$ could only remember the information observed by player $i$ including player $i's$ hidden variable and public actions. Therefore $I_i$ indicates a sequence in IIG, *i.e.*, $x_i^v a_0 a_2 ... a_{L-1}$. For $I_i \in \mathcal{I}_i$ we denote by $A(I_i)$ the set $A(h)$ and by $P(I_i)$ the player $P(h)$ for any $h \in I_i$. For each player $i \in N$ a utility function $u_i(z)$ define the payoff of the terminal state $z$.

For player $i$, the **expected game utility** $u_i^\sigma = \sum_{z \in Z} \pi^\sigma(z) u_i(z)$ of $\sigma$ is the expected payoff of all possible terminal nodes. Given a fixed strategy profile $\sigma_{-i}$, any strategy $\sigma_i^* = \text{argmax}_{\sigma_i' \in \Sigma_i} u_i^{(\sigma_i', \sigma_{-i})}$ of player $i$ that achieves maximize payoff against $\pi_{-i}^\sigma$ is a **best response**. For two players' extensive-form games, a **Nash equilibrium** is a strategy profile $\sigma^* = (\sigma_0^*, \sigma_1^*)$ such that each player's strategy is a best response to the opponent. An $\epsilon$-**Nash equilibrium** is an approximation of a Nash equilibrium, whose strategy profile $\sigma^*$ satisfies: $\forall i \in N, u_i^{\sigma_i^*} + \epsilon \geq \max_{\sigma_i' \in \Sigma_i} u_i^{(\sigma_i', \sigma_{-i})}$. **Exploitability** of a strategy $\sigma_i$ is defined as $\epsilon_i(\sigma_i) = u_i^{\sigma^*} - u_i^{(\sigma_i, \sigma_{-i}^*)}$. A strategy is unexploitable if $\epsilon_i(\sigma_i) = 0$. In large two player zero-sum games such poker, $u_i^{\sigma^*}$ is intractable to compute. However, if the players alternate their positions, the value of a pair of games is zeros, *i.e.*, $u_0^{\sigma^*} + u_1^{\sigma^*} = 0$. We define the exploitability of strategy profile $\sigma$ as $\epsilon(\sigma) = (u_1^{(\sigma_0, \sigma_1^*)} + u_0^{(\sigma_0^*, \sigma_1)})/2$.

### B.2 EXPLANATION BY EXAMPLE

To provide a more detailed explanation, Figure 1 presents an illustration of a partial game tree in One-Card Poker. In the first tree, two players are dealt (queen, jack) as shown in the left subtree and (queen, king) as shown in the right subtree. $z_i$ denotes terminal node and $h_i$ denotes non-terminal node. There are 19 distinct nodes, corresponding 9 non-terminal nodes including chance $h_0$ and 10 terminal nodes in the left tree. The trajectory from the root to each node is a history of actions. In an extensive-form game, $h_i$ refers to this history. For example, $h_3$ consists of actions 0:Q, 1:J and P. $h_7$ consists of actions 0:Q, 1:J, P and B. $h_8$ consists of actions 0:Q, 1:K, P and B. We have $h_3 \sqsubseteq h_7$, $A(h_3) = \{P, B\}$ and $P(h_3) = 1$.

In IIG, the private card of player 1 is invisible to player 0, therefore $h_7$ and $h_8$ are actually the same for player 0. We use infoset to denote the set of these undistinguished states. Similarly, $h_1$ and $h_2$ are in the same infoset. For the right tree of Figure 1, $h_3'$ and $h_5'$ are in the same infoset. $h_4'$ and $h_6'$ are in the same infoset.

Generally, any $I_i \in \mathcal{I}$ could only remember the information observed by player $i$ including player $i's$ hidden variable and public actions. For example, the infoset of $h_7$ and $h_8$ indicates a sequence of 0:Q, P, and B. Because $h_7$ and $h_8$ are undistinguished by player 0 in IIG, all the states have a same strategy. For example, $I_0$ is the infoset of $h_7$ and $h_8$, we have $I_0 = I_0(h_7) = I_0(h_8)$, $\sigma_0(I_0) = \sigma_0(h_7) = \sigma_0(h_8)$, $\sigma_0(a|I_0) = \sigma_0(a|h_7) = \sigma_0(a|h_8)$.

### B.3 Detailed Definition about Strategy and Nash Equilibrium

A **strategy profile** $\sigma = \{\sigma_i | \sigma_i \in \Sigma_i, i \in N\}$ is a collection of strategies for all players, where $\Sigma_i$ is the set of all possible strategies for player $i$. $\sigma_{-i}$ refers to strategy of all players other than player $i$. For play $i \in N$ the **strategy** $\sigma_i(I_i)$ is a function, which assigns an action distribution over $A(I_i)$ to infoset $I_i$. $\sigma_i(a|h)$ denotes the probability of action $a$ taken by player $i \in N \cup \{c\}$ at state $h$. In IIG, $\forall h_1, h_2 \in I_i$, we have $I_i = I_i(h_1) = I_i(h_2)$, $\sigma_i(I_i) = \sigma_i(h_1) = \sigma_i(h_2)$, $\sigma_i(a|I_i) = \sigma_i(a|h_1) = \sigma_i(a|h_2)$. For iterative method such as CFR, $\sigma^t$ refers to the strategy profile at $t$-th iteration.

The **state reach probability** of history $h$ is denoted by $\pi^\sigma(h)$ if players take actions according to $\sigma$. For an empty sequence $\pi^\sigma(\emptyset) = 1$. The reach probability can be decomposed into $\pi^\sigma(h) = \prod_{i \in N \cup \{c\}} \pi_i^\sigma(h) = \pi_i^\sigma(h)\pi_{-i}^\sigma(h)$ according to each player's contribution, where $\pi_i^\sigma(h) = \prod_{h'a \sqsubseteq h, P(h') = P(h)} \sigma_i(a|h')$ and $\pi_{-i}^\sigma(h) = \prod_{h'a \sqsubseteq h, P(h') \neq P(h)} \sigma_{-i}(a|h')$.

The **infoset reach probability** of $I_i$ is defined as $\pi^\sigma(I_i) = \sum_{h \in I_i} \pi^\sigma(h)$. If $h' \sqsubseteq h$, the **interval state reach probability** from state $h'$ to $h$ is defined as $\pi^\sigma(h', h)$, then we have $\pi^\sigma(h', h) = \pi^\sigma(h)/\pi^\sigma(h')$. $\pi_i^\sigma(I_i)$, $\pi_{-i}^\sigma(I_i)$, $\pi_i^\sigma(h', h)$, and $\pi_{-i}^\sigma(h', h)$ are defined similarly.

## C ADDITIONAL EXPERIMENT RESULTS

Figure 8(a) shows that the robust sampling with a larger batch size indicates better performance. It's reasonable because a larger batch size will lead to more sampled infosets in each iteration and costs more memory to store such values. If $b=1$, only one block is sampled in each iteration. The results demonstrate that the larger batch size generally leads to faster convergence. Because it's easy to sample the mini-batch samples by parallel fashion on a large-scale distributed system, this method is very efficient. In practice, we can specify a suitable mini-batch size according to computation and memory size.

In Figure 8(b), we compared the proposed robust sampling against Average Strategy (AS) sampling (Gibson, 2014) on Leduc Hold'em (stack=5). Set the mini-batch size of MCCFR as $b=100$, $k=2$ in robust sampling. The parameters in average strategy sampling are set by $\epsilon=k/|A(I)|$, $\tau=0$, and $\beta=0$. After 1000 iterations, the performance of our robust sampling is better than AS. More specifically, if k=1, the exploitability of our robust sampling is 0.5035 while AS is 0.5781. If k=2, the exploitability of our robust sampling is 0.2791 while AS is 0.3238. Robust sampling samples a $\min(k,|A(I)|)$ player $i$'s actions while AS samples a random number of player $i$'s actions. Note that, if $\rho$ is small or the number of actions is small, it has a possibility that the generated random number between 0 and 1 is larger than $\rho$ for all actions, then the AS will sample zero action. Therefore, AS has a higher variance than our robust sampling. In addition, according to Gibson (2014), the parameter scopes of AS are $\epsilon\in(0,1]$, $\tau\in[1,\infty)$, $\beta\in[0,\infty)$ respectively. They didn't analyze the experiment results for $\tau<1$.

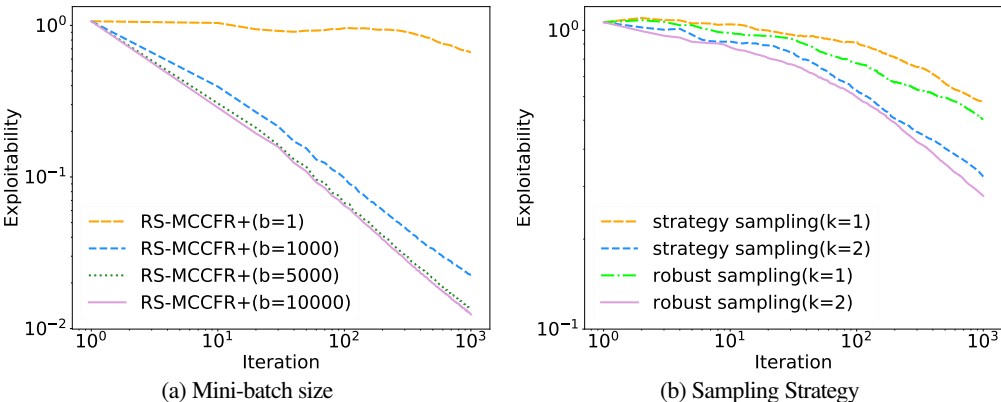

(a) Mini-batch size            (b) Sampling Strategy

Figure 8: Comparison of different CFR-family methods on Leduc Hold'em. (a) Performance of robust sampling with different batch size. (b) Robust sampling vs strategy sampling.

# D  THEORETICAL ANALYSIS

## D.1  REACH PROBABILITY AND POSTERIOR PROBABILITY

**Lemma 2** *The reach probability of the opponent is proportional to posterior probability of the opponent's hidden variable, i.e.,$p(x^v_{-i}|I_i) \propto \pi^\sigma_{-i}(h)$, where $x^v_i$ and $I_i$ indicate a particular $h$.*

**Proof**

For player $i$ at infoset $I_i$ and fixed $i's$ strategy profile $\sigma_i$, i.e., $\forall h \in I_i, \pi^\sigma_i(h)$ is constant. Based on the defination of extensive-form game, the cominbation of $I_i$ and opponent's hidden state $x^v_{-i}$ can indicate a particular history $h = x^v_i x^v_{-i} a_0 a_1 ... a_{L-1}$. With Bayes' Theorem, we can inference the posterior probability of opponent's private cards with Equation9.

$$
\begin{aligned}
p(x^v_{-i}|I_i) &= \frac{p(x^v_{-i}, I_i)}{p(I_i)} = \frac{p(h)}{p(I_i)} \propto p(h) \\
&\propto p(x^v_i) p(x^v_{-i}) \prod_{l=1}^{L} \sigma_{P(x^v_i x^v_{-i} a_0 a_1 ... a_{l-1})}(a_l | x^v_i x^v_{-i} a_0 a_1 ... a_{l-1}) \\
&\propto \pi^\sigma(h) = \pi^\sigma_i(h) \pi^\sigma_{-i}(h) \\
&\propto \pi^\sigma_{-i}(h)
\end{aligned}
\tag{9}
$$

■

## D.2  ROBUST SAMPLING, OUTCOME SAMPLING AND EXTERNAL SAMPLING

For robust sampling, given strategy profile $\sigma$ and the sampled block $Q_j$ according to sampled profile $\sigma^{rs(k)} = (\sigma^{rs(k)}_i, \sigma_{-i})$, then $q(z) = \pi^{\sigma^{rs(k)}}_i(z) \pi^\sigma_{-i}(z)$, and the regret of action $a \in A^{rs(k)}(I_i)$ is

$$
\begin{aligned}
\tilde{r}^\sigma_i((a|I_i)|Q_j) &= \tilde{v}^\sigma_i((a|I_i)|Q_j) - \tilde{v}^\sigma_i(I_i|Q_j) \\
&= \sum_{z \in Q_j, ha \sqsubseteq z, h \in I_i} \frac{1}{q(z)} \pi^\sigma_{-i}(z) \pi^\sigma_i(ha, z) u_i(z) - \sum_{z \in Q_j, h \sqsubseteq z} \frac{1}{q(z)} \pi^\sigma_{-i}(z) \pi^\sigma_i(h, z) u_i(z) \\
&= \sum_{z \in Q_j, ha \sqsubseteq z, h \in I_i} \frac{u_i(z)}{\pi^{\sigma^{rs(k)}}_i(z)} \pi^\sigma_i(ha, z) - \sum_{z \in Q_j, h \sqsubseteq z, h \in I_i} \frac{u_i(z)}{\pi^{\sigma^{rs(k)}}_i(z)} \pi^\sigma_i(h, z) \\
&= \sum_{z \in Q_j, ha \sqsubseteq z, h \in I_i} \pi^\sigma_i(ha, z) u^{rs}_i(z) - \sum_{z \in Q_j, h \sqsubseteq z, h \in I_i} \pi^\sigma_i(h, z) u^{rs}_i(z),
\end{aligned}
\tag{10}
$$

where $u^{rs}_i(z) = \frac{u_i(z)}{\pi^{\sigma^{rs(k)}}_i(z)}$ is the weighted utility according to reach probability $\pi^{\sigma^{rs(k)}}_i(z)$. Because the weighted utility no long requires explicit knowledge of the opponent's strategy, we can use this sampling method for online regret minimization.

Generally, if player $i$ randomly selects $\min(k, |A(I_i)|)$ actions according to discrete uniform distribution $unif(0, |A(I_i)|)$ at infoset $I_i$, i.e., $\sigma^{rs(k)}_i(a|I_i) = \frac{\min(k, |A(I_i)|)}{|A(I_i)|}$, then

$$
\pi^{\sigma^{rs(k)}}_i(I_i) = \prod_{h \in I_i, h' \sqsubseteq h, h'a \sqsubseteq h, h' \in I'_i} \frac{\min(k, |A(I'_i)|)}{|A(I'_i)|}
\tag{11}
$$

and $u^{rs}_i(z)$ is a constant number when given the sampled profile $\sigma^{rs(k)}$.

Specifically,

- if $k = max_{I_i \in I} |A(I_i)|$, then $\sigma^{rs(k)}_i(I_i) = 1$, $u^{rs}_i(z) = u_i(z)$, and

$$\tilde{r}_i^\sigma((a|I_i)|Q_j) = \sum_{z\in Q_j, h\sqsubseteq z, h\in I_i} u_i(z)(\pi_i^\sigma(ha,z) - \pi_i^\sigma(h,z)) \tag{12}$$

Therefore, robust sampling is same with external sampling when $k = max_{I_i\in I}|A(I_i)|$. For large game, because one player should take all actions in her infosets, it's intractable for external sampling. The robust sampling is more flexible and memory-efficient than external sampling. In practice, we can specify a suitable $k$ according our memory. Experimentally, the smaller $k$ can achieve a similar convergence rate to the external sampling.

- if $k = 1$ and $\sigma_i^{rs(k)} = \sigma_i$, only one history $z$ is sampled in this case, then $u_i^{rs(k)}(z) = \frac{u_i(z)}{\pi_i^{\sigma_i}(z)}$, $\exists h \in I_i$, for $a \in A^{rs(k)}(I_i)$

$$\begin{aligned} \tilde{r}_i^\sigma((a|I_i)|Q_j) &= \tilde{r}_i^\sigma((a|h)|Q_j) \\ &= \sum_{z\in Q_j, ha\sqsubseteq z} \pi_i^\sigma(ha,z)u_i^{rs}(z) - \sum_{z\in Q_j, h\sqsubseteq z} \pi_i^\sigma(h,z)u_i^{rs}(z) \\ &= \frac{(1-\sigma_i(a|h))u_i(z)}{\pi_i^\sigma(ha)} \end{aligned} \tag{13}$$

For $a \notin A^{rs(k)}(I_i)$, the regret will be $\tilde{r}_i^\sigma((a|h)|j) = 0 - \tilde{v}_i^\sigma(h|j)$. If we add exploration and guarantee $q(z) \geq \delta > 0$, then robust sampling is same with outcome sampling when $k = 1$ and $\sigma_i^{rs(k)} = \sigma_i$.

- if $k = 1$, and player $i$ randomly selects one action according to discrete uniform distribution $unif(0,|A(I_i)|)$ at infoset $I_i$, then $u_i^{rs(1)}(z) = \frac{u_i(z)}{\pi_i^{\sigma^{rs(k)}}(z)}$ is a constant, $\exists h\in I_i$, for $a\in A^{rs(k)}(I_i)$

$$\begin{aligned} \tilde{r}_i^\sigma((a|I_i)|Q_j) &= \sum_{z\in Q_j, ha\sqsubseteq z, h\in I_i} \pi_i^\sigma(ha,z)u_i^{rs}(z) - \sum_{z\in Q_j, h\sqsubseteq z, h\in I_i} \pi_i^\sigma(h,z)u_i^{rs}(z) \\ &= (1-\sigma_i(a|h))\pi_i^\sigma(ha,z)u_i^{rs(1)}(z) \end{aligned} \tag{14}$$

if action $a$ is not sampled at state $h$, the regret is $\tilde{r}_i^\sigma((a|h)|j) = 0 - \tilde{v}_i^\sigma(h|j)$. Compared to outcome sampling, the robust sampling in that case converges more efficient than outcome sampling. Note that, in our experiment, we select this sampling policy as the default robust sampling when $k = 1$.

## D.3 UNBIASED MINI-BATCH MCCFR

**Theorem 1** $E_{Q^j\sim mini\text{-}batch}[\tilde{v}_i^\sigma(I_i|b)] = v_i^\sigma(I_i)$.

In this section, we prove that mini-Batch MCCFR gives an unbiased estimation of counterfactual value.

**Proof**

$$E_{Q^j \sim \text{mini-batch}}[\tilde{v}_i^\sigma(I_i|b)] = E_{b' \sim \text{unif}(0,b)}[\tilde{v}_i^\sigma(I_i|b')]$$

$$= E_{b' \sim \text{unif}(0,\ b)}\left(\sum_{j=1}^{b'}\sum_{h \in I_i, z \in Q^j, h \sqsubseteq z} \frac{\pi_{-i}^\sigma(z)\pi_i^\sigma(h,z)u_i(z)}{q(z)b'}\right)$$

$$= E_{b' \sim \text{unif}(0,\ b)}\left(\frac{1}{b'}\sum_{j=1}^{b'}\tilde{v}_i^\sigma(I_i|Q^j)\right)$$

$$= \frac{1}{b}\sum_{b'=1}^{b}\left(\frac{1}{b'}\sum_{j=1}^{b'}\tilde{v}_i^\sigma(I_i|Q^j)\right) \tag{15}$$

$$= \frac{1}{b}\sum_{b'=1}^{b}\left(\frac{1}{b'}\sum_{j=1}^{b'}E(\tilde{v}_i^\sigma(I_i|Q^j))\right)$$

$$= \frac{1}{b}\sum_{b'=1}^{b}\left(\frac{1}{b'}\sum_{j=1}^{b'}v_i^\sigma(I_i)\right)$$

$$= v_i^\sigma(I_i)$$

∎

# E  SEQUENTIAL REPRESENTATION AND RECURRENT NEURAL NETWORK WITH ATTENTION

In order to define our $\mathcal{R}$ and $\mathcal{S}$ network, we need to represent the infoset $I_i \in \mathcal{I}$ in extensive-form games. In such games, players take action in an alternating fashion and each player makes a decision according to the observed history. In this paper, we model the behavior sequence as a recurrent neural network and each action in the sequence corresponds to a cell in RNN. Figure 3 (a) provides an illustration of the proposed deep sequential neural network representation for infosets.

In standard RNN, the recurrent cell will have a very simple structure, such as a single *tanh* or *sigmoid* layer. Hochreiter & Schmidhuber (1997) proposed a long short-term memory method (LSTM) with the gating mechanism, which outperforms the standard version and is capable of learning long-term dependencies. Thus we will use LSTM for the representation. Furthermore, different position in the sequence may contribute differently to the decision making, we will add an attention mechanism (Desimone & Duncan, 1995; Cho et al., 2015) to the LSTM architecture to enhance the representation. For example, the player may need to take a more aggressive strategy after beneficial public cards are revealed in a poker game. Thus the information, after the public cards are revealed may be more important.

More specifically, for $l$-th cell, define $x_l$ as the input vector, which can be either player or chance actions. Define $e_l$ as the hidden layer embedding, $\phi_*$ as a general nonlinear function. Each action is represented by a LSTM cell, which has the ability to remove or add information to the cell state with three different gates. Define the notation $\cdot$ as element-wise product. The first **forgetting gate** layer is defined as $g_l^f = \phi_f(w^f[x_l, e_{l-1}])$, where $[x_l, e_{l-1}]$ denotes the concatenation of $x_l$ and $e_{l-1}$. The second **input gate** layer decides which values to update and is defined as $g_l^i = \phi_i(w^i[x_l, e_{l-1}])$. A nonlinear layer outputs a vector of new candidate values $\tilde{C}_l = \phi_c(w^l[x_l, e_{l-1}])$, which decides what can be added to the state. After the forgetting gate and the input gate, the new cell state is updated by $C_l = g_l^f \cdot C_{l-1} + g_l^i \cdot \tilde{C}_l$. The third **output gate** is defined as $g_l^o = \phi_o(w^o[x_l, e_{l-1}])$. Finally, the updated hidden embedding is $e_l = g_l^o \cdot \phi_e(C_l)$. As shown in Figure 3 (a), for each LSTM cell $j$, the vector of attention weight is learned by an **attention network**. Each member in this vector is a scalar $\alpha_j = \phi_a(w^a e_j)$. The attention embedding of $l$-th cell is then defined as $e_l^a = \sum_{j=1}^{l} \alpha_j \cdot e_j$, which is the summation of the hidden embedding $e_j$ and the learned attention weight $\alpha_j$. The final output of the network is predicted by a **value network**, which is defined as

$$\tilde{y}_l := f(a, I_i | \theta) = w^y \phi_v(e_l^a) = w^y \phi_v \left( \sum_{j=1}^{l} \phi_a(w^a e_j) \cdot e_j \right), \tag{16}$$

where $\theta$ refers to the parameters in the defined sequential neural networks. Specifically, $\phi_f$, $\phi_i$, $\phi_o$ are *sigmoid* functions. $\phi_c$ and $\phi_e$ are hyperbolic tangent functions. $\phi_a$ and $\phi_v$ are *rectified linear* functions.

**Remark.** The proposed **R**SN and **A**SN share the same neural architecture, but use different parameters. That is $\mathcal{R}(a, I_i | \theta_\mathcal{R}^t) = f(a, I_i | \theta_\mathcal{R}^t)$ and $\mathcal{S}(a, I_i | \theta_\mathcal{S}^t) = f(a, I_i | \theta_\mathcal{S}^t)$. $\mathcal{R}(\cdot, I_i | \theta_\mathcal{R}^t)$ and $\mathcal{S}(\cdot, I_i | \theta_\mathcal{S}^t)$ denote two vectors of predicted value for all $a \in A(I_i)$.

# F    OPTIMIZING NEURAL REPRESENTATION AND IMPLEMENTATION

## F.1    CODE FOR DNCFR FRAMEWORK

Algorithm 2 provides a summary of the proposed double neural counterfactual regret minimization method. Specifically, in the first iteration, if we start the optimization from tabular-based methods, the techniques in Section 3.4 should be used to clone the cumulative regrets and strategy, which is used to initialize RSN and ASN respectively. If there is no warm start initialization, we can start our algorithm by randomly initializing the parameters in RSN and ASN. After these two kinds of initialization, we use sampling method, such as the proposed robust sampling, to collect the training samples (include infosets and the corresponding values), which are saved in memories $\mathcal{M}_{\mathcal{R}}^t$ and $\mathcal{M}_{\mathcal{S}}^t$ respectively. These samples will be used by the NeuralAgent algorithm from Algorithm 3 to optimize RSN and ASN. Algorithm 4 provides the implementation of the proposed mini-batch robust sampling MCCFR. Note that, with the help of the proposed mini-batch techniques in Section 4, we can collect training samples parallelly on multi-processors or distributed systems, which also leads to the unbiased estimation according to the proved Theorem 1. The acceleration training and distribution implementation is beyond the scope of this paper. To compare the performance of DNCFR and tabular CFR, all of our experiments are running on a single processor.

## F.2    CODE FOR NEURAL NETWORKS

---

**Algorithm 3:** Optimization of Deep Neural Network

---

**Function** NeuralAgent($f(\cdot|\theta^{T-1})$, $\mathcal{M}$, $\theta^{T-1}$, $\beta^*$):

    initialize $optimizer$, $scheduler$

    $\theta^T \leftarrow \theta^{T-1}$, $l_{best} \leftarrow \infty$, $t_{best} \leftarrow 0$

    **For** $t=1$ $to$ $\beta_{epoch}$ **do**

        $loss \leftarrow []$

        **For** $each\ training\ epoch$ **do**

            $\{x^{(i)},y^{(i)}\}_{i=1}^m \sim \mathcal{M}$

            $batch\_loss \leftarrow \frac{1}{m}\sum_{i=1}^m (f(x^{(i)}|\theta^{T-1}) + y^{(i)} - f(x^{(i)}|\theta^T))^2$

            back propagation $batch\_loss$ with learning rate $lr$

            clip gradient of $\theta^T$ to $[-\epsilon,\epsilon]^d$

            $optimizer(batch\_loss)$

            $loss.append(batch\_loss)$

        $lr \leftarrow sheduler(lr)$

        **if** $avg(loss) < \beta_{loss}$ **then**

            $\theta_{best}^T \leftarrow \theta^T$, early stopping.

        **else if** $avg(loss) < l_{best}$ **then**

            $l_{best} = avg(loss)$, $t_{best} \leftarrow t$, $\theta_{best}^T \leftarrow \theta^T$

        **if** $t - t_{best} > \beta_{re}$ **then**

            $lr \leftarrow \beta_{lr}$

    **return** $\theta^T$

---

**Notations in Neural Networks.** Define $\beta_{epoch}$ as training epoch, $\beta_{lr}$ as learning rate, $\beta_{loss}$ as the criteria for early stopping, $\beta_{re}$ as the upper bound for the number of iterations from getting the minimal loss last time, $\theta^{t-1}$ as the old parameter learned in $t-1$ iteration, $f(\cdot|\theta^{t-1})$ as the neural network, $\mathcal{M}$ as the training samples including infosets and the corresponding targets. To simplify notations, we use $\beta^*$ to denote the set of hyperparameters in the proposed deep neural networks. $\beta_{\mathcal{R}}^*$ and $\beta_{\mathcal{S}}^*$ refer to the sets of hyperparameters in RSN and ASN respectively.

**Optimize Neural Networks.** Algorithm 3 provides the implementation of the optimization technique for both RSN and ASN. Both $\mathcal{R}(a,I_i|\theta_{\mathcal{R}}^{t+1})$ and $\mathcal{S}(a,I_i|\theta_{\mathcal{S}}^t)$ are optimized by mini-batch stochastic gradient descent method. In this paper, we use Adam optimizer (Kingma & Ba, 2014) with both momentum and adaptive learning rate techniques. We also replace Adam by other optimizers such as Nadam, RMSprop, Nadam Ruder (2017) in our experiments, however, such optimizers do not achieve better experimental results. In practice, existing optimizers (Ruder, 2017) may not return a relatively low enough loss because of potential saddle points or local minima. To obtain a relatively higher accuracy and lower optimization loss, we design a novel scheduler to reduce the learning rate when the loss has stopped decrease. Specifically,

the scheduler reads a metrics quantity, $e.g$, mean squared error. If no improvement is seen for a number of epochs, the learning rate is reduced by a factor. In addition, we will reset the learning rate in both optimizer and scheduler once loss stops decreasing within $\beta_{re}$ epochs. Gradient clipping mechanism is used to limit the magnitude of the parameter gradient and make optimizer behave better in the vicinity of steep cliffs. After each epoch, the best parameters, which lead to the minimum loss, will replace the old parameters. Early stopping mechanism is used once the lowest loss is less than the specified criteria $\beta_{loss}$.

**The feature is encoded as following.** As shown in the figure 3 (a), for a history $h$ and player $P(h)$, we use vectors to represent the observed actions including chance player. For example, on Leduc Hold'em, the input feature $x_l$ for $l$-th cell is the concatenation of three one-hot features including the given private cards, the revealed public cards and current action $a$. Both the private cards and public cards are encoded by one-hot technique (Harris & Harris), where the value in the existing position is 1 and the others are 0. If there are no public cards, the respective position will be filled with 0. The betting chips in the encoded vector will be represented by the normalized cumulative spent, which is the cumulative chips dividing the stack size. For HUNL, each card is encoded by a vector with length 17: 13 for ranking embedding and 4 for suit embedding. The actions in public sequences are represented by one-hot and the raise action is also represented by the normalized cumulative spent.

## F.3 Code for Mini-Batch Robust Sampling MCCFR

---

**Algorithm 4:** Mini-Batch RS-MCCFR with Double Neural Networks

---

**Function** `Mini-Batch-MCCFR-NN`$(t)$**:**

    $\mathcal{M}_{\mathcal{R}}^t \leftarrow \emptyset, \mathcal{M}_{\mathcal{S}}^t \leftarrow \emptyset$

    **For all** $i=1$ *to* $b$ **do in parallel then**

        MCCFR-NN$(t,\emptyset,0,1,1)$

        MCCFR-NN$(t,\emptyset,1,1,1)$

    **return** $\mathcal{M}_{\mathcal{R}}^t, \mathcal{M}_{\mathcal{S}}^t$

 

**Function** `MCCFR-NN`$(t, h, i, \pi_i, \pi_i^{rs(k)})$**:**

    $I_i \leftarrow I_i(h)$

    **if** $h \in Z$ **then**

        **return** $\frac{u_i(h)}{\pi_i^{rs(k)}}$

    **else if** $P(h)=-1$ **then**

        $a \sim \sigma_{-i}(I_i)$

        **return** MCCFR-NN$(t,ha,i,\pi_i,\pi_i^{rs(k)})$

    **else if** $P(h)=i$ **then**

        $\hat{R}_i(\cdot|I_i) \leftarrow \mathcal{R}(\cdot,I_i|\theta_{\mathcal{R}}^t)$ **if** $t>1$ **else** $\overrightarrow{0}$

        $\sigma_i(I_i) \leftarrow$ CalculateStrategy$(\hat{R}_i(\cdot|I_i),I_i)$

        $v_i(h) \leftarrow 0, r_i(\cdot|I_i) \leftarrow \vec{0}, s_i(\cdot|I_i) \leftarrow \vec{0}$

        $A^{rs(k)}(I_i) \leftarrow$ sampling $k$ different actions according to $\sigma_i^{rs(k)}$

        **For** $a \in A^{rs(k)}(I_i)$ **do**

            $v_i(a|h) \leftarrow$ MCCFR-NN$(t,ha,i,\pi_i\sigma_i(a|I_i),\pi_i^{rs}\sigma_i^{rs(k)}(a|I_i))$

            $v_i(h) \leftarrow v_i(h)+v_i(a|h)\sigma_i(a|I_i)$

        **For** $a \in A^{rs(k)}(I_i)$ **do**

            $r_i(a|I_i) \leftarrow v_i(a|h)-v_i(h)$

            $s_i(a|I_i) \leftarrow \pi_i^\sigma(I_i)\sigma_i(a|I_i)$

        Store updated cumulative regret tuple $(I_i,r_i(\cdot|I_i))$ in $\mathcal{M}_{\mathcal{R}}^t$

        Store updated current strategy dictionary $(I_i,s_i(\cdot|I_i))$ in $\mathcal{M}_{\mathcal{S}}^t$

        **return** $v_i(h)$

    **else**

        $\hat{R}_{-i}(\cdot|I_i) \leftarrow \mathcal{R}(\cdot,I_i|\theta_{\mathcal{R}}^t)$ **if** $t>1$ **else** $\overrightarrow{0}$

        $\sigma_{-i}(I_i) \leftarrow$ CalculateStrategy$(\hat{R}_{-i}(\cdot|I_i),I_i)$

        $a \sim \sigma_{-i}(I_i)$

        **return** MCCFR-NN$(t,ha,i,\pi_i,\pi_i^{rs(k)})$

**Function** `CalculateStrategy`$(R_i(\cdot|I_i),I_i)$**:**

    $sum \leftarrow \sum_{a \in A(I_i)} \max(R_i(a|I_i),0)$

    **For** $a \in A(I_i)$ **do**

        $\sigma_i(a|I_i) = \frac{\max(R_i(a|I_i),0)}{sum}$ **if** $sum > 0$ **else** $\frac{1}{|A(I_i)|}$

    **return** $\sigma_i(I_i)$

---

Algorithm 4 presents one application scenario of the proposed mini-batch robust sampling method. The function MCCFR-NN will traverse the game tree like tabular MCCFR, which starts from the root $h=\emptyset$. Define $I_i$ as the infoset of $h$. Suppose that player $i$ will sample $k$ actions according to the robust sampling. Algorithm 4 is defined as follows.

●If the history is terminal, the function returns the weighted utility.

●If the history is the chance player, one action $a \in A(I_i)$ will be sampled according to the strategy $\sigma_{-i}(I_i)$. Then this action will be added to the history, *i.e.*, $h \leftarrow ha$.

•If $P(I_i) = i$, the current strategy can be updated by the cumulative regret predicted by RSN. Then we sample $k$ actions according the specified sampled strategy profile $\sigma_i^{rs(k)}$. After a recursive updating, we can obtain the counterfactual value and regret of each action at $I_i$. For the observed nodes, their counterfactual regrets and numerators of the corresponding average strategy will be stored in $\mathcal{M}_{\mathcal{R}}^t$ and $\mathcal{M}_{\mathcal{S}}^t$ respectively.

•If $P(I_i)$ is the opponent, only one action will be sampled according the strategy $\sigma_{-i}(I_i)$.

The function Mini-Batch-MCCFR-NN presents a mini-batch sampling method, where $b$ blocks will be sampled in parallel. This mini-batch method can help the MCCFR to achieve an unbiased estimation of CFV. The parallel implementation makes this method efficient in practice.

**Remark:** We update average in the procedure of $P(h) = i$, which potentially leads to a biased estimate of average strategy. There is a trade-off among unbiased estimate, convergence, and data efficiency on Algorithm 4. A feasible solution is using stochastically-weighted averaging (SWA). However, SWA typically leads to a large variance as discussed in Marc's Ph.D. thesis (Lanctot, 2013) (p49). The classical external sampling(ES) solves this problem by only updating average strategy for $-i$. Because ES samples $k = |A(I_i)|$ actions for $i$ and only samples one action for $-i$, it's inefficient to collect samples for average strategy at $-i$ in neural CFR. In contrast, we collect samples at $i$. Typically, when collecting average strategy samples at $i$, we need using SWA to maintain unbiased estimate of average strategy. However, because of the high variance of SWA, we find the one without SWA converges more efficient empirically.

### F.4 HYPERPARAMETERS

In experiments, we set the network hyperparameters as following.

**Hyperparameters on Leduc Hold'em**. The Leduc(5), Leduc(10) and Leduc(15) in our experiments have $1.1 \times 10^4$ infosets ($6 \times 10^4$ states), $3 \times 10^5$ ($1.5 \times 10^6$ states) and $3 \times 10^6$ ($2 \times 10^7$ states) infosets respectively. We set $k = 3$ as the default parameter in the provable robust sampling method on all such games. For the small Leduc(5), we select $b = 100$ as the default parameter in the mini-batch MCCFR **??**, which only samples $5.59\%$ infosets in each iteration. For the larger Leduc(10) and Leduc(15), we select default $b = 500$, which visit (observe) only $2.39\%$ and $0.53\%$ infosets in each iteration. To train RSN and ASN, we set the default embedding size for both neural networks as 16, 32, and 64 for Leduc(5), Leduc(10), and Leduc(15) respectively. There are 256 samples will be used to update the gradients of parameters by mini-batch stochastic gradient descent technique. We select Adam (Kingma & Ba, 2014) as the default optimizer and LSTM with attention as the default neural architecture in all the experiments. The neural networks only have 2608, 7424, and 23360 parameters respectively, which are much less than the number of infosets. The default learning rate of Adam is $\beta_{lr} = 0.001$. A scheduler, who will reduce the learning rate based on the number of epochs and the convergence rate of loss, help the neural agent to obtain a high accuracy. The learning rate will be reduced by 0.5 when loss has stopped improving after 10 epochs. The lower bound on the learning rate of all parameters in this scheduler is $10^{-6}$. To avoid the algorithm converging to potential local minima or saddle points, we will reset the learning rate to 0.001 and help the optimizer to obtain a better performance. $\theta_{best}^T$ is the best parameters to achieve the lowest loss after $T$ epochs. If average loss for epoch $t$ is less than the specified criteria $\beta_{loss} = 10^{-4}$ for RSN (set this parameter as $10^{-5}$ for RSN), we will early stop the optimizer. We set $\beta_{epoch} = 2000$ and update the optimizer 2000 maximum epochs. For ASN, we set the loss of early stopping criteria as $10^{-5}$. The learning rate will be reduced by 0.7 when loss has stopped improving after 15 epochs.

For NFSP in our experiment, we set the hyperparameters according to its original paper (Heinrich & Silver, 2016). The neural network in NFSP had 1 hidden layer of 64 neurons and rectified linear activation. The reinforcement and supervised learning rates were set to 0.1 and 0.005. Both neural networks were optimized by vanilla stochastic gradient descent for every 128 steps in the game. The mini-batch sizes for both neural networks were 128. The sizes of memories were 200k and 2m for reinforcement learning and supervised learning respectively. we set the anticipatory parameter in NFSP to 0.1. The exploration in $\epsilon$-greedy policies started at 0.06 and decayed to 0.

**Hyperparameters on HUNL**. To solve HUNL(1) and HUNL(2), we sample $0.01\%$ and $0.001\%$ infosets in each iteration. The batch size of training neural network is set to 100000. We prefer to using large batch size, because gradient descent spends most of running time. Typically, larger batch size indicates less number of gradient decent updates. We perform DNCFR under different number of embedding sizes and the steps of gradient descent updates. The experiment results are presented in Figure 7. Other hyperparameters in neural networks and optimizers are set to be the same with Leduc(15).

# G    ABS-CFR, DEEPSTACK, DOUBLE NEURAL CFR AND HUNL

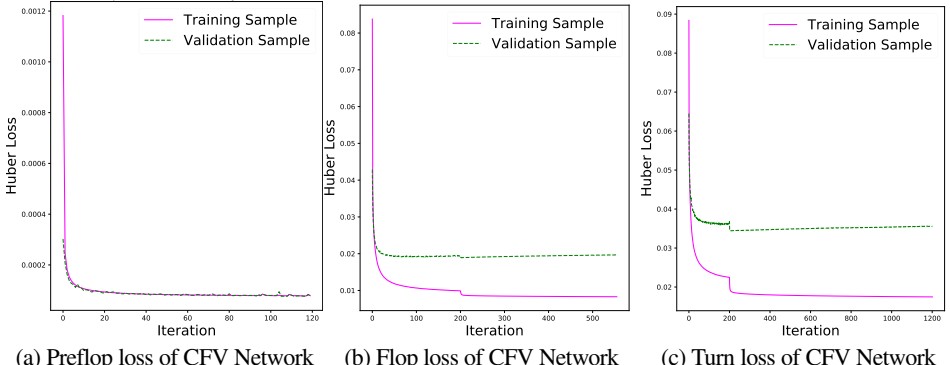

(a) Preflop loss of CFV Network    (b) Flop loss of CFV Network    (c) Turn loss of CFV Network

Figure 9: Huber loss of three counterfactual value network in our implemented DeepStack. (a) Huber loss of auxiliary network on preflop subgame, the training loss is 0.0000789 and the validation loss is 0.0000776 while they are 0.000053 and 0.000055 in original DeepStack. (b) Huber loss of deep counterfactual value network on flop subgame, the training sample is 0.008 and the validation sample is 0.019 while they are 0.008 and 0.034 in original DeepStack. (c) Huber loss of deep counterfactual value network on turn subgame (contains last two rounds of HUNL), the training sample is 0.016 and the validation sample is 0.035 while they are 0.016 and 0.026 in original DeepStack. Specifically, the learning rate is decayed in 200th iteration(iteration is equal to epoch here), therefore the huber loss in (b) and (c) decreased. To balance the performance of both training and validation samples, we finally select the checkpoints that have the lowest validation loss.

The game size of imperfect information HUNL is compared with Go (Silver et al., 2016) and her partial observable property makes it very difficult. The article (Burch, 2017) gives a detailed analysis of this problem from the perspective of both computational time and space complexity. To evaluate the proposed method, we reimplement DeepStack (Moravcik et al., 2017), which is an expert-level artificial intelligence in Heads-up No-Limit Texas Hold'em. DeepStack defeated professional poker players. The decision points of Heads-up No-Limit Texas Hold'em exceed $10^{161}$ (Johanson, 2013). We provide the game rules of Texas hold'em in Appendix A.3.

In this section, we provided some details about our implementation, compared our agent with the original DeepStack to guarantee the correctness of the implementation, and applied our double neural method on the subgame of DeepStack.

## G.1    DETAILS ABOUT ABS-CFR

ABS-CFR agent is an enhanced version of HITSZ_LMW_2pn, whose previous version won the third prize of the 2018 Annual Computer Poker Competition (ACPC) and has $2 \times 10^{10}$ information sets. The ideas of ABS-CFR agent is first abstract the full HUNL into the smaller abstract game and using CFR to solve the abstracted game. The ABS-CFR using two kind-of abstractions: the first one is action abstraction and the second is card abstraction. The action abstraction is using discretized betting model (Gilpin et al., 2008), which can do fold, call, $0.5\times$ pot raise, $1\times$ pot raise, $2\times$ pot raise, and $4\times$ pot raise and all-in in each decision node. The card abstraction is using domain knowledge that strategically similar states are collapsed into a single state. In preflop we use lossless abstraction which has 169 buckets. In flop and turn, we use potential-aware imperfect-recall abstraction with earth mover distance (Ganzfried & Sandholm, 2014), which has 10000 and 50000 buckets respectively. In the river, we use opponent cluster hand strength abstraction (Johanson et al., 2013), which has 5000 buckets.

## G.2    DETAILS ABOUT OUR IMPLEMENTATION OF DEEPSTACK

Because Alberta university didn't release the source code of DeepStack for No-Limit Texas Hold'em, we implemented this algorithm according to the original article (Moravcik et al., 2017). It should be noted

that the released example code [4] on Leduc Hold'em cannot directly be used on Heads-up No-Limit Texas Hold'em for at least three reasons: (1) The tony game Leduc Hold'em only has 2 rounds, 6 cards with default stack size 5, which is running on a single desktop, while HUNL has four rounds, 52 cards and stack size 20000 according to ACPC game rules. (2) Specifically, there are 55,627,620,048,000 possible public and private card combinations for two players on HUNL (Johanson, 2013) and the whole game contains about $10^{161}$ infosets, which makes the program should be implemented and run on a large-scale distributed computing cluster. (3) The example code doesn't contain the necessary acceleration techniques and parallel algorithm for Texas Hold'em.

Our implementation follows the key ideas presented in the original DeepStack article by using the same hyperparameters and training samples. To optimize the counterfactual value network on turn subgame (this subgame looks ahead two rounds and contains both turn and river), we generate nine million samples. Because each sample is generated by traversing 1000 iterations using CFR+ algorithm based on a random reach probability, these huge samples are computation-expensive and cost 1500 nodes cluster (each node contains 32 CPU cores and 60GB memory) more than 60 days. To optimize the counterfactual value network on flop subgame (this subgame only looks ahead one round), we generate two million samples, which costs about one week by using the similar computation resource. The auxiliary network on preflop subgame is optimized based on ten million samples and costs 2 days. The whole implementation of DeepStack costs us several months and hundreds of thousands of lines of codes.

### G.3 VERIFY THE CORRECTNESS OF OUR IMPLEMENTATION

The overall DeepStack algorithm contains three ingredients: (1) computing strategy for the current public state, (2) depth-limited Lookahead to the end of subgame rather than the end of the full game and using counterfactual value network to inference the value of the leaf node in the subgame, (3) using action abstraction technique to reduce the size of game tree.

To evaluate the strategy of imperfect information game, exploitability is usually used as the metric to evaluate the distance between the strategy and Nash equilibrium in two-player zero-sum game. However, in the large game, such as Heads-Up No-Limit Texas Hold'em, computation of exploitability is expensive because of its $10^{161}$ searching space.

We verified the correctness of our implementation from three different aspects: First, the logs of DeepStack against professional poker players are released on the official website, which contains more than 40000 hand histories. From these logs, we counted the frequency of each action taken by DeepStack under different private cards and used the normalized frequency as the estimated strategy of DeepStack. We compared this estimated strategy with our reimplemented DeepStack. Figure 10 in Appendix G provided the comparison results and demonstrated that our implementation leads to policies very close to what the original DeepStack does. Second, we compared the huber loss of three deep counterfactual value networks. Clearly, our implementation achieved a loss similar to the original paper. Third, our agent also played against an enhanced version of HITSZ_LMW_2pn, whose previous version won the third prize of the 2018 Annual Computer Poker Competition (ACPC). Our implementation can win HITSZ_LMW_2pn 120 mbb/g.

---

[4] https://github.com/lifrordi/DeepStack-Leduc

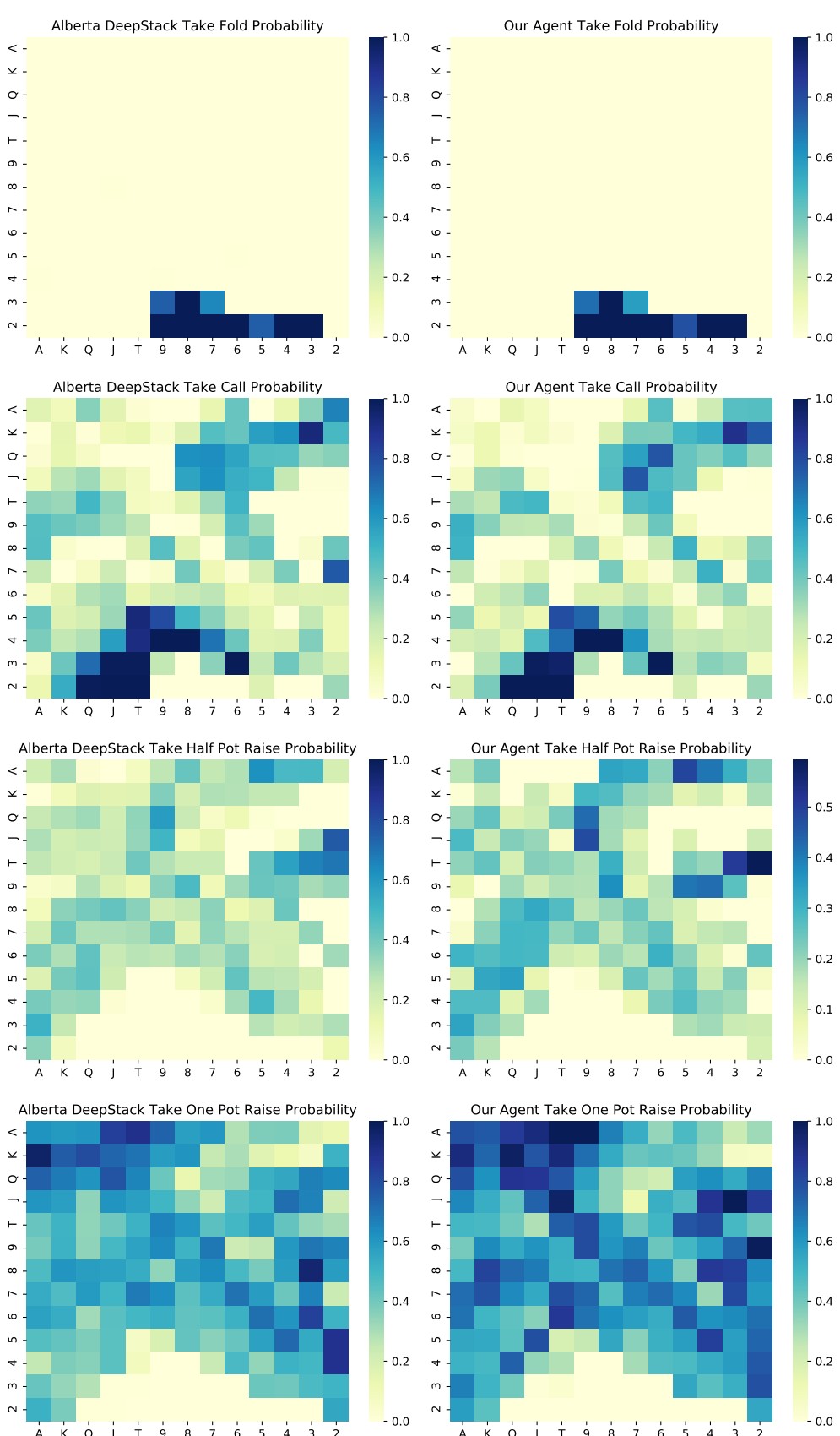

Figure 10: Comparison of action probability between Alberta's DeepStack (Moravcik et al., 2017) (the left column) and our reimplemented DeepStack (the right column).

