# OpenReview forum: "Double Neural Counterfactual Regret Minimization"
_ICLR.cc/2020/Conference — Accept (Poster)_

### Official Review · AnonReviewer3 · 2019-10-23
**Official Blind Review #3**

**Rating:** 6

**Review:**

The authors proposed a double neural counterfactual regret minimization algorithm (DNCFR) that uses a RegretSumNetwork to approximate cumulative regret and an AvgStrategyNetwork to approximate the average strategy. To help the training, the authors use robust sampling and a mini-batch training methods. The main contributions of the paper are: First, the DNCFR algorithm and the recurrent neural network architecture; Second, an efficient sampling and training method; Third, plenty of experiments that corroborate the effectiveness of DNCFR.

It is interesting and meaningful to develop neural-based CFR, in order to eliminate the manual abstraction and apply CFR to large-scale imperfect information games. The authors tested their algorithm on a medium scale game HUNL(1) (with 2 * 10 ^ 8 information sets) and trained a blueprint strategy on large scale game HUNL(2), which is combined with value networks from DeepStack and beats ABS-CFR a lot. It is great to see that DNCFR works on large scale games. However, both HUNL(1) and HUNL(2) are one-round games and it not clear how to combine the blueprint strategy trained by DNCFR with DeepStack. What’s more, as DNCFR is only effective on first round as the blueprint strategy trainer when played against ABS-CFR, it is more likely that DeepStack beats ABS-CFR, instead of DNCFR beats it. So the result in Figure 7(c) is not so convincing.

Unlike tabular CFR that save regrets and strategies for all the information sets or other neural-based algorithms that need large reservoir buffers. It only needs to save data sampled from the most recent iterations, which saves much memory. In fact, this is a bootstrap method borrowed from Reinforcement learning. Though the method save memory and has lower variance than methods that use reservoir buffers, it is bias as it trains the new RSN and ASN based on the output of the old networks. It seems good when the game size is small and the CFR iterations is small. It may needs very large CFR batches and very many gradient descent updates when training on large scale games, in order to control the bias. The results in Figure 7(a) and 7(b) are limited in CFR iterations. Experiments using different gradient descent updates and different CFR batch while given more CFR iterations should be tested, in order to show the effect of the bias training.
In “Algorithm 4”. The calculation of average strategy seems wrong. Because you are using MCCFR, According to “Monte Carlo sampling and regret minimization for equilibrium computation and decision-making in large extensive form games”, you may need a method call “stochastically-weighted averaging”. It should be noted that the sampling probability of
each information set is not equal. You may need to discuss this.

The authors train the network for 2000 updates when the batch size is 256 for Leduc and 100000 for HUNL(1) and HUNL(2) in every CFR iteration (I am not sure how much gradient updates are used in HUNL(2), it is not given). There's quite a lot of updates in every CFR iteration. But it is acceptable when compared to Deep CFR proposed by Brown, which uses 4000 updates and the batch size is 10000.

Experiments:
1. In the ablation studies, the algorithms are tested on small scale game Leduc(5). It is quite a small game that event the size neural parameters is larger than the size of information sets. It is OK but larger games make more sense. Especially in the
experiment of “Individual network”, as this experiment is important to show that
DNCFR is comparable to tabular CFR and the bias is acceptable.
2. The paper didn’t show what the learned regret and average strategy looks. If they are
showed, it would be helpful to understand the bias in the bootstrap learning.
3. In the part “Is robust sampling helpful”, the authors want to show that the robust sampling with k=1 is better than outcome sampling. But I didn’t find how they set the exploration parameter in outcome sampling and I am afraid that it doesn’t make sense. Because outcome sampling has a parameter to adjust the exploration. According to "Monte Carlo sampling and regret minimization for equilibrium computation and decision-making in large extensive form games", the best exploration parameter is different in different game, but it is almost sure that totally exploration is not the best setting (it is equivalent to the robust sampling with k = 1).
4. In the part “Do the neural networks generalize to unseen infosets”. The authors claims that it is true. But the experiment only shows that the neural network don’t forget
information sets that trained before.
5. In the part “How well does DNCFR on larger games”, the DNCFR is limited to 100
iterations while is allow to run for 1000 iterations in other experiments. 100 iterations
are too few to show the effectiveness of DNCFR on these games.
6. The algorithm is tested on HUNL(1) and HUNL(2), which are one round and action- abstracted version of HUNL. But the authors should give more detail description of
these games.
7. It is not clear how to combine the blueprint strategy trained by DNCFR with
DeepStack, as DeepStack uses continual resolving and don’t need any blueprint strategy. And it would be interesting if the head-to-head performance of DNCFR agent on large scale games (for example, the FHP with two rounds and more than 1e^9 information sets) is reported, instead of the performance of the agent that combined with DeepStack.
8. In section 5.4, “When variance reduction techniques are applied, Figure 7(c)...”. The authors didn’t explain why the variance reduction techniques are needed here, but in order to compare the algorithm directly, some other advanced techniques should not be used here.

**Experience Assessment:**

I have read many papers in this area.

**Review Assessment: Checking Correctness Of Derivations And Theory:**

N/A

**Review Assessment: Checking Correctness Of Experiments:**

I assessed the sensibility of the experiments.

**Review Assessment: Thoroughness In Paper Reading:**

I read the paper at least twice and used my best judgement in assessing the paper.

---

> ### Author Response · Authors · 2019-11-15
> **To Reviewer 3**
>
> Thanks for your positive and constructive comments. Hopefully, this reply can address all your concerns.
>
> 1. HUNL
>
> 1.1 variance reduction:
> It’s well known that head-to-head evaluation of HUNL is challenging because of its high variance. We use AIVAT[1] to reduce evaluation variance. AIVAT is also used in DeepStack and Pluribus.
>
> 1.2 Setting and evaluation:
> We train blueprint strategies on a finer-grained abstracted HUNL(2), which contains about $8*10^10$ infosets. Its terminal values are estimated by the flop network rather than auxiliary network. After that, we employ continue resolving to compute strategy in real-time for the next rounds.
>
> To make a fair comparison, in Fig.7(c), we show two head-to-head competitions: DNCFR against ABS-CFR and tabular agent. Note that, DeepStack’s continue resolving is intractable to solve HUNL(2) in “real-time”. Specifically, DeepStack needs GPU (16GB) to accelerate continue resolving, however, large games will result in out of memory. DeepStack on CPU cannot obtain strategy in dozens of seconds per action. Therefore, tabular agent is a reasonable benchmark via replacing continue resolving by tabular blueprint strategy (saving in a large table). We show that the neural agent obtains a similar win rate with the tabular agent while using hundreds of times less memory.
>
> We believe this setting is convincing to demonstrate the advantages of DNCFR against tabular CFR.
>
> 1.3 long-running performance:
> It’s very expensive to compute exploitability in large games. Till the submission deadline, we only collected the results within hundreds of iterations (similar to Noam’s Deep CFR). Now, we report the further convergence on 1k iterations as follows. Typically, larger iterations and SGD updates will return better strategies.
>
> Fig.7(a): the settings under embedding size 8, 16, 32, 64, 128 approach $27.48\pm0.07, 26.88\pm0.03,17.71\pm0.03, 8.92\pm0.03, 6.52\pm0.02$.
>
> Fig.7(b): the settings under SGD updates approach $16.85\pm0.51, 6.52\pm0.02, 1.88\pm0.07$
>
> 2. average strategy:
> Good question. That’s a trade-off among unbiased estimate, convergence, and data efficiency. Stochastically-weighted averaging (SWA) typically leads to a large variance as discussed in Marc’s Ph.D. thesis (p49). The classical external sampling(ES) solves this problem by only updating average strategy for $-i$. Because ES samples $k=|A(I_i)|$ actions for $i$ and only samples one action for $-i$, it’s inefficient to collect samples for average strategy at $-i$ in neural CFR. In contrast, we collect samples at $i$. Typically, when collecting average strategy samples at $i$, we need using SWA to maintain unbiased estimate of average strategy as you said. However, because of the high variance of SWA, we find the one without SWA converges more efficient empirically. Specifically, we test these methods on Leduc. After 1k iterations, exploitability of the method with/without SWA are 0.43 and 0.14 (SWA is worse because of high variance).
>
> 3. Bias in neural network:
> Yes. It has a bias when training neural networks. Analysis of this bias is interesting and challenging, and out of the scope for this paper. Furthermore, we compare the final average strategy of tabular MCCFR and DNCFR on HUNL(1), the MSE and KL-divergence are 0.003 and 0.024 respectively (very small difference). Also, we recognize it’s just an approximate solution to analyze the overall bias because IIGs could have many Nash equilibria.
>
> 4. generalization to unseen infoset:
> We learn new parameters based on the old parameters and a subset of observed samples. All infosets share the same parameters, so that the neural network can estimate the values for unseen infosets. Note that, #parameters is orders of magnitude less than #infosets in many settings, which indicates the generalization of our method. Furthermore, Fig.4(d) shows that DNCFRs are slightly better than tabular MCCFR, we think it’s because of the generalization to unseen infosets.
>
> 5. sampling method:
> We set exploration as 1/t. Outcome sampling (OS) with total exploration is uniform sampling rather than robust sampling(RS) with k=1. In Sec.4.1 and Appendix.D.2, we introduce that RS is a general version of both OS and ES. In OS, each player samples one action according to her policy with exploration. For question “Is robust sampling helpful”, RS with k=1 refers to the traverser uniformly samples one action while the opponent samples one action according to her policy. Furthermore, concurrent work[2] also demonstrates that RS(k=1) is more efficient than OS and uniform sampling.
>
> Reference:
> [1] Neil Burch et.al. AIVAT: A New Variance Reduction Technique for Agent Evaluation in Imperfect Information Games. 2016.
> [2] Trevor Davis et.al. Low-Variance and Zero-Variance Baselines for Extensive-Form Games. 2019.

---

### Official Review · AnonReviewer2 · 2019-10-24
**Official Blind Review #2**

**Rating:** 8

**Review:**

The paper introduces two neural networks, one for average regrets and one for average strategy, to (approximately) run CFR algorithm in (large) IIG. New sampling techniques are used to train these networks to represent the resulting strategy.

I remember seeing this paper in the last year's OpenReview for ICLR - I was not reviewing the paper.
I enjoyed the first version of the paper, but it was missing experiments on larger games and some questions on smaller games were unanswered.
In this version, authors clearly spent a large amount of time (including re-implementing DeepStack!) so that they could compare on large games (namely HU no-limit Poker) and overall greatly improved the paper and evaluation.

The evaluation on small games includes comparison to NFSP/XFP, as well as investigating time/space trade-off.
For the large game, I like that the authors evaluated against an ACPC agent.
Previous work is well cited, and authors have a good overall map of related work (both older results and new papers).

Issues:

1) One downside of the paper is that it is very close to the "Deep Counterfactual Regret Minimization".
While authors devote a full paragraph in section 6 to contrast these, the difference is relatively small.
I do not think it is fair to dwell too much on this though, since the first version of the paper with this idea originally came *before* DeepCFR publication!

2) Since the approach is so similar to DeepCFR, it would be nice to include it in comparison (not just NFSP/XFP).


Minor details:

- Page 9: "...which has no limited number of actions, ..." rephrase please, this sounds like the game is infinite.

- Page 9: ", more abstracted action leads to better strategy..." more abstracted sounds like it is smaller, rephrase please to something like "finer grained abstraction".

- Minor frequent grammatical issues, but does not derail from the flow and semantics of the paper.

Conclusion:

Overall, the paper introduces method that is interesting to the community, scales to large games and the paper includes comprehensive evaluation section.
I believe it should be accepted.


**Experience Assessment:**

I have published in this field for several years.

**Review Assessment: Checking Correctness Of Derivations And Theory:**

I assessed the sensibility of the derivations and theory.

**Review Assessment: Checking Correctness Of Experiments:**

I assessed the sensibility of the experiments.

**Review Assessment: Thoroughness In Paper Reading:**

I read the paper at least twice and used my best judgement in assessing the paper.

---

> ### Author Response · Authors · 2019-11-15
> **To Reviewer 2**
>
> Thanks for your recognition of our work.
>
> Exactly, we have made a lot of efforts to evaluate our method on the large-scale game (heads-up no-limit Texas Hold’em) in the past year. In the future, we plan to reproduce the latest works, such as ED, Deep CFR, Single Deep CFR, and compare them on HUNL, although it's expected that such evaluation needs a lot of work and computation resources.

---

### Decision · Program_Chairs · 2019-12-19

**Decision:**

Accept (Poster)

**Comment:**

Double coúnterfactual regret minimization is an extension of neural counterfactual regret minimization that uses separate policy and regret networks (reminiscent of similar extensions of the basic RL formula in reinforcement learning). Several new algorithmic modifications are added to improve the performance.

The reviewers agree that this paper is novel, sound, and interesting. One of the reviewers had a set of questions that the authors responded to, seemingly satisfactorily. Given that this seems to be a high-quality paper with no obvious issues, it should be accepted.